# A CTP-dependent gating mechanism enables ParB spreading on DNA

Adam SB Jalal[1], Ngat T Tran[1], Clare EM Stevenson[2], Afroze Chimthanawala[3,4], Anjana Badrinarayanan[3], David M Lawson[2], Tung BK Le[1]*

[1]Department of Molecular Microbiology, John Innes Centre, Norwich, United Kingdom; [2]Department of Biochemistry and Metabolism, John Innes Centre, Norwich, United Kingdom; [3]National Centre for Biological Sciences, Tata Institute of Fundamental Research, Bangalore, India; [4]SASTRA University, Thanjavur, Tamil Nadu, India

**ABSTRACT** Proper chromosome segregation is essential in all living organisms. The ParA-ParB-*parS* system is widely employed for chromosome segregation in bacteria. Previously, we showed that *Caulobacter crescentus* ParB requires cytidine triphosphate to escape the nucleation site *parS* and spread by sliding to the neighboring DNA (Jalal et al., 2020). Here, we provide the structural basis for this transition from nucleation to spreading by solving co-crystal structures of a C-terminal domain truncated *C. crescentus* ParB with *parS* and with a CTP analog. Nucleating ParB is an open clamp, in which *parS* is captured at the DNA-binding domain (the DNA-gate). Upon binding CTP, the N-terminal domain (NTD) self-dimerizes to close the NTD-gate of the clamp. The DNA-gate also closes, thus driving *parS* into a compartment between the DNA-gate and the C-terminal domain. CTP hydrolysis and/or the release of hydrolytic products are likely associated with reopening of the gates to release DNA and recycle ParB. Overall, we suggest a CTP-operated gating mechanism that regulates ParB nucleation, spreading, and recycling.

*For correspondence:
tung.le@jic.ac.uk

Competing interest: The authors declare that no competing interests exist.

## Introduction

Proper chromosome segregation is essential in all domains of life. In most bacterial species, faithful chromosome segregation is mediated by the tripartite ParA-ParB-*parS* system (*Donczew et al., 2016*; *Fogel and Waldor, 2006*; *Harms et al., 2013*; *Ireton et al., 1994*; *Jakimowicz et al., 2002*; *Jalal and Le, 2020a*; *Kawalek et al., 2018*; *Lin and Grossman, 1998*; *Livny et al., 2007*; *Mohl et al., 2001*; *Tran et al., 2018*). ParB, a CTPase and DNA-binding protein, nucleates on *parS* before spreading to adjacent non-specific DNA to form a higher-order nucleoprotein complex (*Breier and Grossman, 2007*; *Broedersz et al., 2014*; *Graham et al., 2014*; *Jalal and Le, 2020a*; *Murray et al., 2006*; *Rodionov et al., 1999*; *Sanchez et al., 2015*; *Taylor et al., 2015*). The ParB-DNA nucleoprotein complex stimulates the ATPase activity of ParA, driving the movement of the *parS* locus (and subsequently, the whole chromosome) to the opposite pole of the cell (*Hwang et al., 2013*; *Leonard et al., 2005*; *Lim et al., 2014*; *Taylor et al., 2021*; *Vecchiarelli et al., 2014*; *Vecchiarelli et al., 2012*). ParB spreads by sliding along the DNA, in a manner that depends on the binding of a co-factor, cytidine triphosphate (CTP) (*Balaguer F de et al., 2021*; *Jalal et al., 2020c*; *Osorio-Valeriano et al., 2019*; *Soh et al., 2019*). A co-crystal structure of a C-terminal domain truncated *Bacillus subtilis* ParB (ParBΔCTD) together with CDP showed the nucleotide to be sandwiched between adjacent subunits, thus promoting their dimerization (*Soh et al., 2019*). A similar arrangement was seen in the co-crystal structure of an N-terminal domain (NTD) truncated version of the *Myxococcus xanthus* ParB homolog, PadC, bound to CTP (*Osorio-Valeriano et al., 2019*). Self-dimerization at the NTD of *B. subtilis* ParB creates a clamp-like molecule that enables DNA entrapment (*Soh et al., 2019*). Biochemical studies

with *M. xanthus* and *C. crescentus* ParBs showed that CTP facilitates the dissociation of ParB from *parS*, thereby switching ParB from a nucleating mode to a sliding mode (*Jalal et al., 2020c*; *Osorio-Valeriano et al., 2019*). ParB can hydrolyze CTP to CDP and inorganic phosphate (*Jalal et al., 2020c*; *Osorio-Valeriano et al., 2019*; *Soh et al., 2019*); however, hydrolysis is not required for spreading since ParB in complex with a non-hydrolyzable CTP analog (CTPɣS) can still self-load and slide on DNA (*Jalal et al., 2020c*; *Soh et al., 2019*). Furthermore, *M. xanthus* PadC does not possess noticeable CTPase activity (*Osorio-Valeriano et al., 2019*). As such, the role of CTP hydrolysis in bacterial chromosome segregation is not yet clear.

Here, we solve co-crystal structures of a C-terminal domain truncated *C. crescentus* ParB in complex with either *parS* or CTPɣS to better understand the roles of CTP binding and hydrolysis. Consistent with the previous report (*Soh et al., 2019*), the NTDs of *C. crescentus* ParB also self-dimerize upon binding to nucleotides, thus closing a molecular gate at this domain (the NTD-gate). Furthermore, the two opposite DNA-binding domains (DBD) move closer together to close a second molecular gate (the DNA-gate). We provide evidence that the CTP-induced closure of the DNA-gate drives *parS* DNA from the DBD into a 20-amino-acid long compartment between the DNA-gate and the C-terminal domain, thus explaining how CTP binding enables ParB to escape the high-affinity *parS* site to spread while still entrapping DNA. Lastly, we identify and characterize a ParB 'clamp-locked' mutant that is defective in CTP hydrolysis but otherwise competent in gate closing, suggesting a possible role for CTP hydrolysis/release of hydrolytic products in the reopening ParB gates and in recycling ParB. Collectively, we suggest a CTP-operated gating mechanism that might regulate ParB nucleation, spreading, and recycling.

## Results

### Co-crystal structure of a *C. crescentus* ParBΔCTD-*parS* complex reveals an open conformation at the NTD

We sought to solve a co-crystal structure of *C. crescentus* ParB nucleating at *parS*. After screening several constructs with different lengths of ParB and *parS*, we obtained crystals of a 50 amino acid C-terminally truncated ParB in complex with a 22 bp *parS* DNA (*Figure 1*). This protein variant lacks the C-terminal domain (CTD) responsible for ParB dimerization (*Figure 1A*; *Figge et al., 2003*). Diffraction data for the ParBΔCTD-*parS* co-crystal were collected to 2.9 Å resolution, and the structure was solved by molecular replacement (see Materials and methods). The asymmetric unit contains four copies of ParBΔCTD and two copies of the *parS* DNA (*Figure 1—figure supplement 1A,B*).

Each ParBΔCTD subunit consists of an NTD (helices α1–α4 and sheets β1–β4) and a DBD (helices α5–α10) (*Figure 1B*). Each ParBΔCTD binds to a half *parS* site, but there is no protein-protein contact between the two adjacent subunits (*Figure 1B*). We previously reported a 2.4 Å co-crystal structure of the DBD of *C. crescentus* ParB bound to *parS* (*Jalal et al., 2020b*) and elucidated the molecular basis for specific *parS* recognition, hence we focus on the conformation of the NTD here instead. We observed that helices α3 and α4 are packed towards the DBD and are connected to the rest of the NTD via an α3–β4 loop (*Figure 1B,C*). While the DBD and helices α3–α4 are near identical between the two ParBΔCTD subunits (root-mean-square deviation [RMSD] = 0.19 Å, *Figure 1C*), the rest of the NTD, from α1 to β4, adopts notably different conformations in the two subunits (*Figure 1C,D*). Specifically, NTDs (α1–β4) from the two ParBΔCTD subunits are related by a rotation of approximately 80° due to changes in a flexible loop in between α3 and β4 (*Figure 1D*). Furthermore, by superimposing the *C. crescentus* ParBΔCTD-*parS* structure onto that of *Helicobacter pylori* (*Chen et al., 2015*), we observed that the NTDs of ParB from both species can adopt multiple alternative orientations (*Figure 1—figure supplement 2*). Taken together, these observations suggest that the ability of the NTD to adopt multiple open conformations is likely a general feature of nucleating ParB.

### Co-crystal structure of a *C. crescentus* ParBΔCTD-CTPɣS complex reveals a closed conformation at the NTD

Next, to gain insight into the spreading state of ParB, we solved a 2.7 Å resolution structure of *C. crescentus* ParBΔCTD in complex with CTPɣS (see Materials and methods). At this resolution, it was not possible to assign the position of the ligand's sulfur atom. Indeed, the placement of the sulfur atom relative to the terminal phosphorus atom may vary from one ligand to the next in the crystal,

leading to an averaging of the electron density. Hence, we modeled CTP, instead of CTPγS, into the electron density (*Figure 2* and *Figure 2—figure supplement 1*). The asymmetric unit contains two copies of ParBΔCTD, each with a CTPγS molecule and a coordinated $Mg^{2+}$ ion bound at the NTD (*Figure 2A*). In contrast to the open conformation of the ParBΔCTD-*parS* structure, nucleotide-bound NTDs from opposite subunits self-dimerize (with an interface area of 2111 $Å^2$, as determined by PISA; *Krissinel, 2015*), thus adopting a closed conformation (*Figure 2A*). Multiple CTPγS-contacting residues also directly contribute to the NTD self-dimerization interface (summarized in *Figure 2—figure*

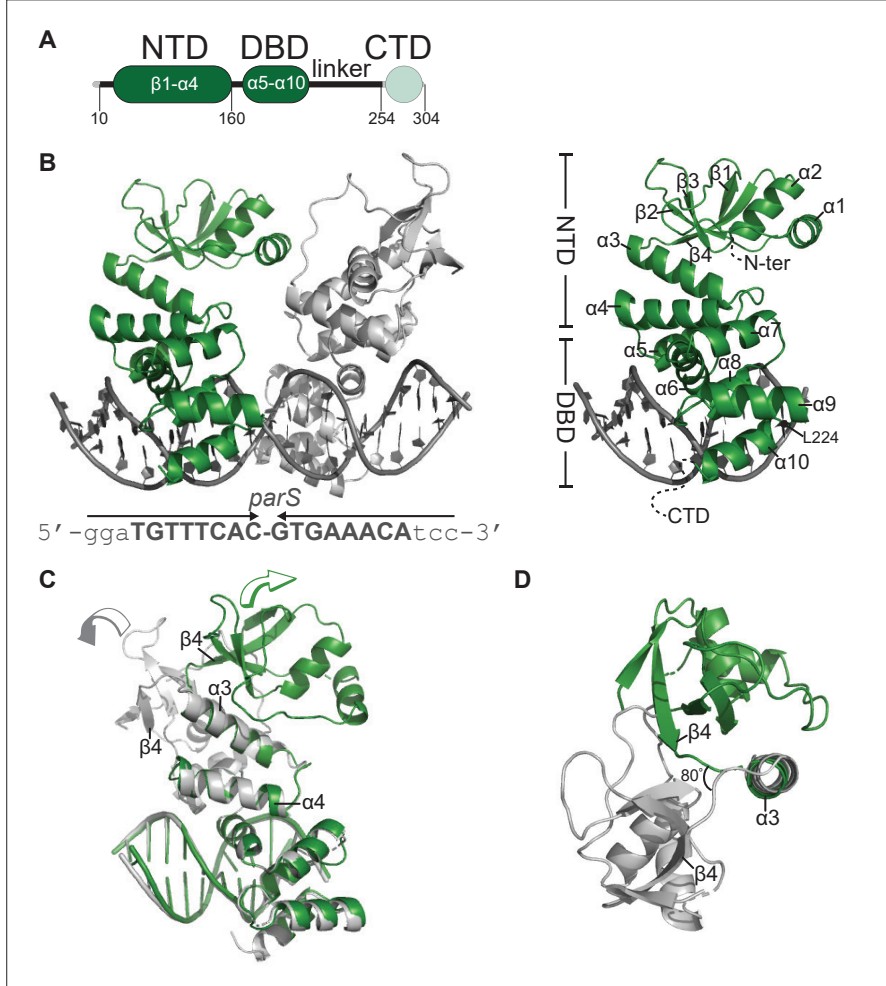

**Figure 1.** Co-crystal structure of a *C. crescentus* ParBΔCTD-*parS* complex reveals an open conformation at the N-terminal domain (NTD). (**A**) The domain architecture of *C. crescentus* ParB: the NTD (dark green), the central DNA-binding domain (DBD, dark green), the C-terminal domain (CTD, faded green), and a linker that connects the DBD and the CTD together. The ParBΔCTD variant that was used for crystallization lacks the CTD (faded green). (**B**, left panel) Co-crystal structure of two *C. crescentus* ParBΔCTD monomers (dark green and gray) bound to a 22 bp *parS* DNA. The nucleotide sequence of the 22 bp *parS* is shown below the co-crystal structure, the core *parS* sequence is highlighted in bold, and each *parS* half-site is denoted by an arrow. The position of residue L224 is also indicated. (Right panel) The structure of a ParBΔCTD subunit bound to a *parS* half site with key features highlighted. (**C**) Superimposition of *C. crescentus* ParBΔCTD subunits shows two different orientations of the NTD. The arrow above each subunit shows the direction each NTD is projecting towards. (**D**) A top-down view of the superimposition of ParBΔCTD subunits shows their NTDs orienting ~80° apart from each other.

The online version of this article includes the following figure supplement(s) for figure 1:

**Figure supplement 1.** The composition of the asymmetric unit (ASU) of the *C. crescentus* ParBΔCTD-*parS* co-crystal.

**Figure supplement 2.** Structural comparisons of the *C. crescentus* ParBΔCTD-*parS* complex to the *H. pylori* ParBΔCTD-*parS* complex.

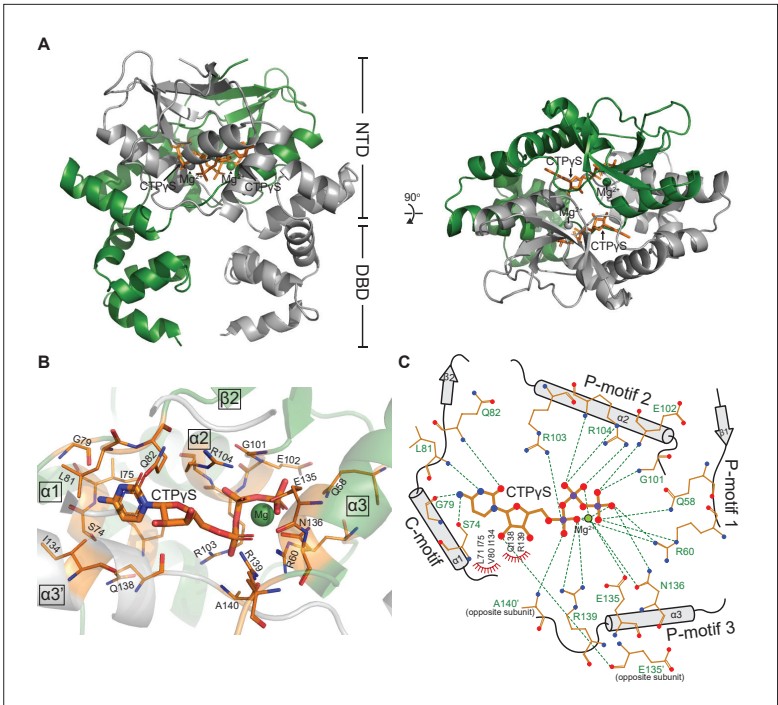

**Figure 2.** Co-crystal structure of a *C. crescentus* ParBΔCTD-CTPγS complex reveals a closed conformation at the N-terminal domain (NTD). (**A**, left panel) The front view of the co-crystal structure of *C. crescentus* ParBΔCTD (dark green and gray) bound to a non-hydrolyzable analog CTPγS (orange) and Mg$^{2+}$ ions (dark green and gray spheres). (Right panel) The top view of the *C. crescentus* ParBΔCTD-CTPγS co-crystal structure. Note that helix α10 is not resolved in this structure due to a poor electron density in this region. (**B**) The nucleotide-binding pocket of *C. crescentus* ParB showing amino acid residues that contact the CTPγS molecule and the coordinated Mg$^{2+}$ ion. (**C**) Protein-ligand interaction map of CTPγS bound to *C. crescentus* ParBΔCTD. Hydrogen bonds are shown as dashed green lines and hydrophobic interactions as red semi-circles. Nitrogen, oxygen, phosphate, and magnesium atoms are shown as blue, red, purple, and green filled circles, respectively.

The online version of this article includes the following figure supplement(s) for figure 2:

**Figure supplement 1.** Omit mF$_{obs}$-DF$_{calc}$ difference electron density calculated at 2.73 Å resolution for Mg-CTP.

**Figure supplement 2.** Sequence alignment of the chromosomal ParB protein family.

**Figure supplement 3.** Structural comparisons of the *C. crescentus* ParBΔCTD-CTPγS complex to the *B. subtilis* ParBΔCTD-CDP complex and the *M. xanthus* PadCΔNTD-CTP complex.

*supplement 2*), indicating a coupling between nucleotide binding and self-dimerization. Furthermore, the *C. crescentus* ParBΔCTD-CTPγS structure is similar to that of the CDP-bound *B. subtilis* ParBΔCTD (RMSD = 1.48 Å) and the CTP-bound *M. xanthus* PadCΔNTD (RMSD = 2.23 Å) (*Figure 2—figure supplement 3A*), suggesting that the closed conformation at the NTD is structurally conserved in nucleotide-bound ParB/ParB-like proteins.

Each CTPγS molecule is sandwiched between helices α1, α2, α3 from one subunit and helix α3' from the opposite subunit (*Figure 2B*). Ten amino acids form hydrogen-bonding contacts with three phosphate groups of CTPγS, either directly or via the coordinated Mg$^{2+}$ ion (*Figure 2C*). These phosphate-contacting residues are referred to as P-motifs 1–3, respectively (P for phosphate motif, *Figure 2C*). Four amino acids at helix α1 and the α1–β2 intervening loop provide hydrogen-bonding interactions to the cytosine ring, hence are termed the C-motif (C for cytosine motif, *Figure 2C*). Lastly, six additional residues contact the ribose moiety and/or the pyrimidine moiety via hydrophobic interactions (*Figure 2C*). Nucleotide-contacting residues in *C. crescentus* ParB and their corresponding amino acids in ParB/ParB-like homologs are summarized in *Figure 2—figure supplement 2* and *Figure 2—figure supplement 3B*. The C-motif forms a snug fit to the pyrimidine moiety, thus is incompatible with larger purine moieties such as those from ATP or GTP. Hydrogen-bonding contacts from the G79 main chain and the S74 side chain to the amino group at position 4 of the cytosine moiety further

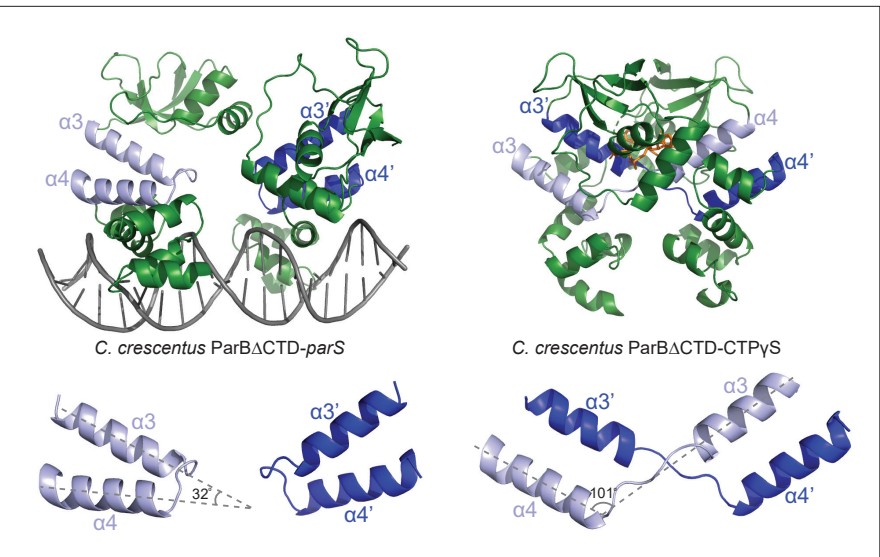

**Figure 3.** Conformational changes between the nucleating and the spreading states of *C. crescentus* ParB. Structures of *C. crescentus* ParBΔCTD in complex with *parS* (left panel) and with CTPγS (right panel), with the pairs of helices (α3–α4, and α3'–α4' for the opposite subunit) shown in light blue and dark blue, respectively. Below each structure, only the α3–α4, α3'–α4' pairs, and the angles between these helices are shown.

distinguish CTP from UTP (*Figure 2C*). Taken all together, our structural data are consistent with the known specificity of *C. crescentus* ParB for CTP (*Jalal et al., 2020c*).

## Conformational changes between the nucleating and the spreading state of *C. crescentus* ParB

A direct comparison of the *C. crescentus* ParBΔCTD-*parS* structure to the ParBΔCTD-CTPγS structure further revealed the conformational changes upon nucleotide binding. In the nucleating state, as represented by the ParBΔCTD-*parS* structure, helices α3 and α4 from each subunit bundle together (32° angle between α3 and α4, *Figure 3*). However, in the spreading state, as represented by the ParBΔCTD-CTPγS structure, α3 swings outwards by 101° to pack itself with α4' from the opposing subunit (*Figure 3*). Nucleotide binding most likely facilitates this 'swinging-out' conformation since both α3 and the α3–α4 loop, that is, P-motif 3 make numerous contacts with the bound CTPγS and the coordinated $Mg^{2+}$ ion (*Figure 2C*). The reciprocal exchange of helices ensures that the packing in the α3–α4 protein core remains intact, while likely driving the conformational changes for the rest of the NTD as well as the DBD (*Figure 4A*). Indeed, residues 44–121 at the NTD rotate wholesale by 94° to dimerize with their counterpart from the opposing subunit (*Figure 4A* and *Figure 4—figure supplement 1A*). Also, residues 161–221 at the DBD rotate upward by 26° in a near rigid-body movement (*Figure 4A* and *Figure 4—figure supplement 1A*). As a result, the opposite DBDs are closer together in the spreading state (inter-domain distance = ~ 27 Å) than in the nucleating state (inter-domain distance = ~ 36 Å) (*Figure 4—figure supplement 1B*). By overlaying the CTPγS-bound structure onto the *parS* DNA complex, it is clear that the DBDs in the spreading state clash severely with DNA, hence are no longer compatible with *parS* DNA binding (*Figure 4B*). Our structural data are therefore consistent with the previous finding that CTP decreases *C. crescentus* ParB nucleation on *parS* or liberates pre-bound ParB from *parS* site (*Jalal et al., 2020c*). Overall, we suggest that CTP binding stabilizes a conformation that is incompatible with DNA-binding and that this change might facilitate ParB escape from the high-affinity nucleation *parS* site.

## *C. crescentus* ParB entraps *parS* DNA in a compartment between the DBD and the CTD in a CTP-dependent manner

To verify the CTP-dependent closed conformation of ParB, we performed site-specific crosslinking of purified proteins using a sulfhydryl-to-sulfhydryl crosslinker bismaleimidoethane (BMOE) (*Soh et al., 2019*). Residues Q35, L224, and I304 at the NTD, DBD, and CTD, respectively (*Figure 5A*), were

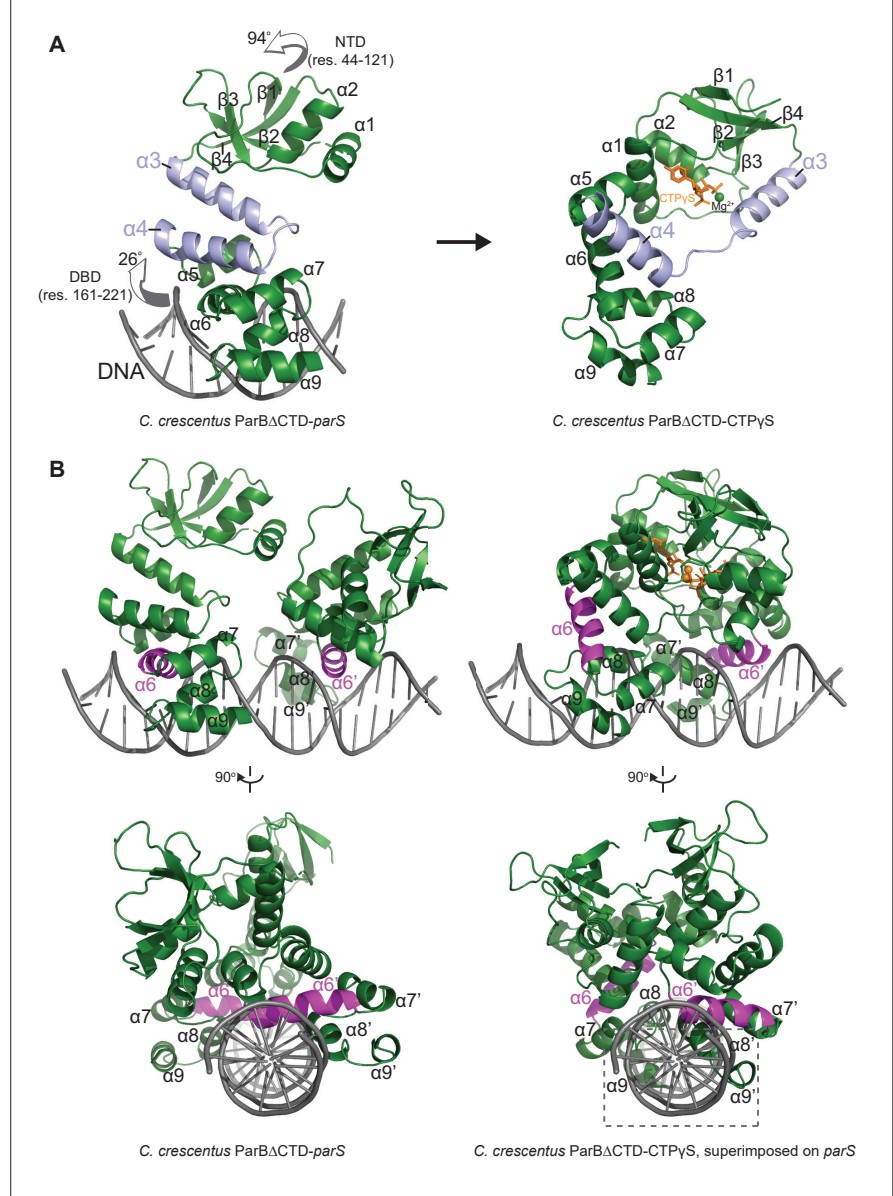

**Figure 4.** The structure of a nucleotide-bound *C. crescentus* ParBΔCTD is incompatible with specific *parS* binding at the DNA-binding domain (DBD). (**A**) Structural changes between *C. crescentus* ParBΔCTD-*parS* and ParBΔCTD-CTPγS structures. Helices α3 and α4 are shown in light blue. The arrows next to the N-terminal domain (NTD) (residues 44–121) and the DBD (residues 161–221) show the direction that these domains rotate towards in the nucleotide-bound state. (**B**) Superimposing the *C. crescentus* ParBΔCTD-CTPγS structure onto *parS* DNA shows DNA-recognition helices (α6 and α6', magenta) positioning away from the two consecutive major grooves of *parS*, and helices α8–α9 and α8'–α9' at the DBD (dashed box) clashing with *parS* DNA.

The online version of this article includes the following figure supplement(s) for figure 4:

**Figure supplement 1.** The structure of nucleotide-bound *C. crescentus* ParBΔCTD is incompatible with specific *parS* binding at the DNA-binding domain (DBD).

substituted individually to cysteine on an otherwise cysteine-less ParB (C297S) background (*Jalal et al., 2020c*), to create ParB variants where symmetry-related cysteines become covalently linked if they are within 8 Å of each other (*Figure 5B*). We observed that the crosslinking of both ParB (Q35C) and ParB (L224C) was enhanced ~2.5–3-fold in the presence of *parS* DNA and CTP (*Figure 5B*), consistent with CTP favoring a conformation when the NTD and the DBD are close together. In contrast,

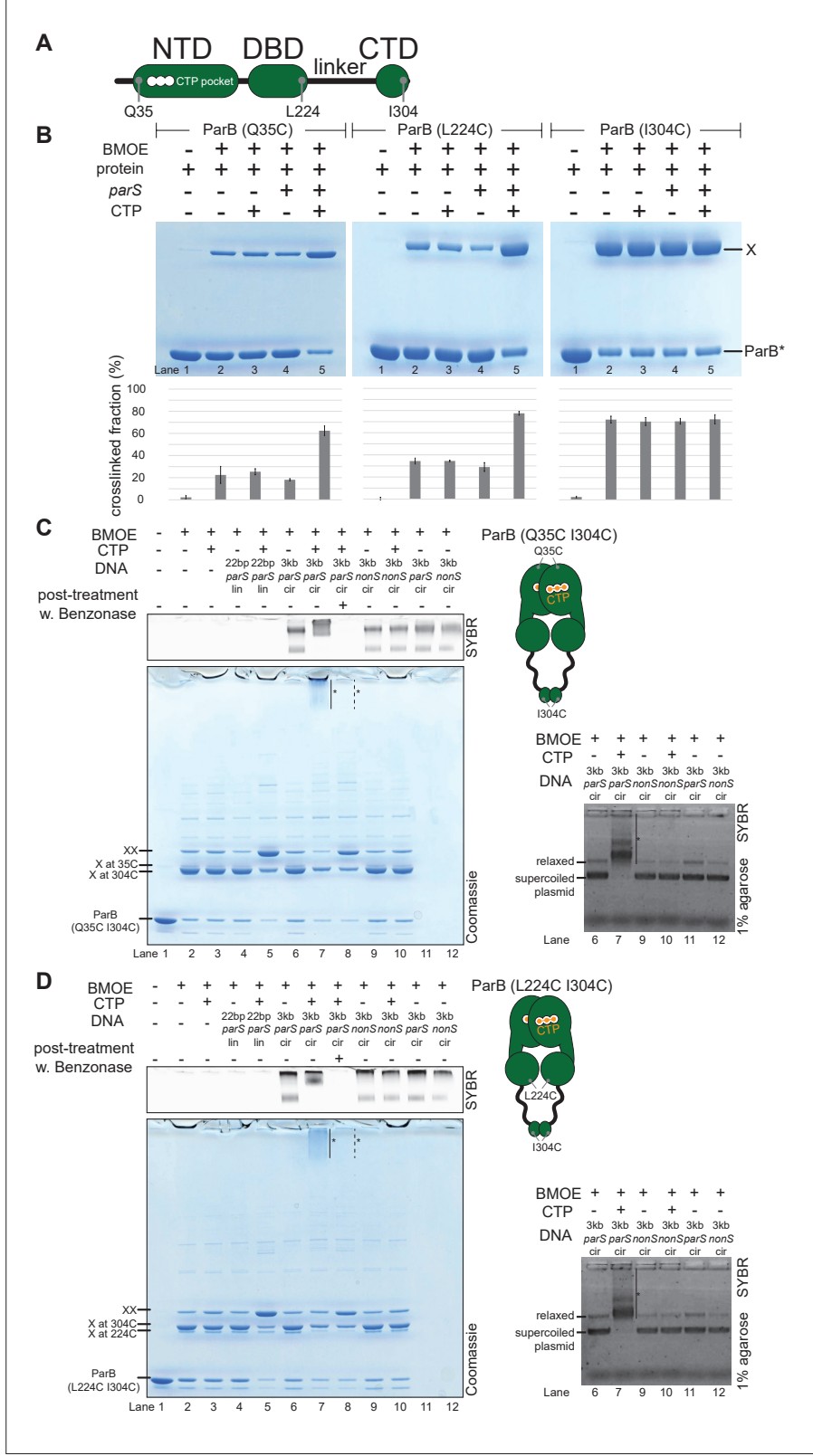

**Figure 5.** *C. crescentus* ParB entraps *parS* DNA in a compartment between the DNA-binding domain (DBD) and the C-terminal domain (CTD) in a cytidine triphosphate (CTP)-dependent manner. (**A**) A schematic diagram of *C. crescentus* ParB showing the position of Q35 (at the N-terminal domain [NTD]), L224 (at the DBD), and I304 (at the CTD) that were substituted either individually or in combinations for cysteine. (**B**) Denaturing polyacrylamide

*Figure 5 continued on next page*

*Figure 5 continued*

gel analysis of bismaleimidoethane (BMOE) crosslinking products of 8 µM single-cysteine ParB (Q35C/L224C/I304C) variant ±0.5 µM 22 bp *parS* DNA ±1 mM CTP. X indicates a crosslinked form of ParB. Quantification of the crosslinked (X) fraction is shown below each representative gel image. Error bars represent SD from three replicates. (**C**, left panel) Denaturing polyacrylamide gel analysis of BMOE crosslinking products of 8 µM dual-cysteine ParB (Q35C I304C) variant ±0.5 µM DNA ±1 mM CTP. Different DNA were employed in crosslinking reactions: a linear 22 bp *parS* DNA (22 bp *parS* lin), a circular 3 kb *parS* plasmid (3 kb *parS* cir), and a circular 3 kb scrambled *parS* plasmid (3 kb *nonS* cir). The high molecular weight (HMW) smear near the top of the polyacrylamide gel is marked with a solid line and an asterisk (lane 7). When the crosslinking reaction was post-treated with a non-specific DNA nuclease, Benzonase, the HMW smear was no longer observed (dashed line and asterisk, lane 8). The polyacrylamide gel was also stained with a DNA dye, Sybr Green (SYBR), and only the top section of the gel is shown. Small 22 bp *parS* DNA duplex migrated out of the gel, thus was not observed near the top of the Sybr-stained gel. A schematic diagram of a dual-cysteine *C. crescentus* ParB dimer is also shown. (Right panel) Agarose gel analysis of BMOE crosslinking products. A subset of crosslinking reactions (lanes 6, 7, and 9–12) were loaded and resolved on 1 % agarose gel. The gel was subsequently stained with Sybr Green for DNA. Shifted gel bands are marked with a solid line and an asterisk. (**D**) Same as panel (**C**) but another dual-cysteine variant, ParB (L224C I304C) was employed instead.

The online version of this article includes the following figure supplement(s) for figure 5:

**Source data 1.** Original files, annotation of the full raw gels, and data used to generate **Figure 5**.

**Figure supplement 1.** Crosslinking ParB (Q35C I304C) and ParB (L224C I304C) did not produce a high molecular weight (HMW) smear in the presence of cytidine triphosphate (CTP) and a linearized *parS* plasmid.

**Figure supplement 1—source data 1.** Original files, annotation of the full raw gels, and data used to generate *Figure 5—figure supplement 1*.

**Figure supplement 2.** Crosslinking ParB (Q35C L224C) did not produce a high molecular weight (HMW) smear despite the presence of cytidine triphosphate (CTP) and a circular *parS* plasmid.

**Figure supplement 2—source data 1.** Original files, annotation of the full raw gels, and data used to generate *Figure 5—figure supplement 2*.

**Figure supplement 3.** The high molecular weight (HMW) smear likely contains catenates between crosslinked ParB dimers and circular *parS* plasmids.

**Figure supplement 3—source data 1.** Original files, annotation of the full raw gels, and data used to generate *Figure 5—figure supplement 3*.

**Figure supplement 4.** A premature closing of ParB clamps prevents their interactions with a 170 bp closed *parS* DNA substrate.

**Figure supplement 4—source data 1.** Data used to generate *Figure 5—figure supplement 4*.

ParB (I304C) crosslinked independently of CTP or *parS* (*Figure 5B*), supporting the known role of the CTD as a primary dimerization domain (*Figge et al., 2003*; *Fisher et al., 2017*).

Previously, it was shown that *B. subtilis* ParB-CTP forms a protein clamp that entraps DNA (*Soh et al., 2019*); however, the location of DNA within the clamp is not yet clear. To locate such DNA-entrapping compartment, we employed a double crosslinking assay while taking advantage of the availability of crosslinkable cysteine residues in all three domains of *C. crescentus* ParB (*Figure 5A*). A *C. crescentus* ParB variant with crosslinkable NTD and CTD interfaces (Q35C I304C) was first constructed and purified (*Figure 5C*). ParB (Q35C I304C) could form high molecular weight (HMW) species near the top of the polyacrylamide gel in the presence of CTP, a 3 kb *parS* plasmid, and the crosslinker BMOE (lane 7, *Figure 5C*, left panel). The HMW smear on the polyacrylamide gel contained both protein and DNA as apparent from a dual staining with Coomassie and Sybr Green (*Figure 5C*, left panel). Slowly migrating DNA-stained bands were also observed when resolved on an agarose gel (*Figure 5C*, right panel). The HMW smear most likely contained DNA-protein catenates between a circular *parS* plasmid and a denatured but otherwise circularly crosslinked ParB (Q35C I304C) polypeptide. Indeed, a post-crosslinking treatment with Benzonase, a non-specific DNA nuclease (lane 8, *Figure 5C*, left panel) or the use of a linearized *parS* plasmid (lane 2 vs. lane 4, *Figure 5—figure supplement 1*) eliminated the HMW smear, presumably by unlinking the DNA-protein catenates. Lastly, the HMW smear was not observed when a plasmid containing a scrambled *parS* site was used (lane 10, *Figure 5C*, left panel) or when CTP was omitted from the crosslinking reaction (lane 6, *Figure 5C*, left panel), indicating that the DNA entrapment is dependent on *parS* and CTP. Collectively, these experiments demonstrate

that as with the *B. subtilis* ParB homolog, *C. crescentus* ParB is also a CTP-dependent molecular clamp that can entrap *parS* DNA in between the NTD and the CTD.

Employing the same strategy, we further narrowed down the DNA-entrapping compartment by constructing a ParB (L224C I304C) variant in which both the DBD and the CTD are crosslinkable (*Figure 5D*). We found that crosslinked ParB (L224C I304C) also entrapped circular plasmid efficiently in a *parS*- and CTP-dependent manner, as judged by the appearance of the HMW smear near the top of the gel (lane 7, *Figure 5D*, left panel). By contrast, ParB (Q35C L224C) that has both the NTD and the DBD crosslinkable was unable to entrap DNA in any tested condition (*Figure 5—figure supplement 2*). We therefore hypothesized that ParB clamps entrap DNA within a compartment created by a 20-amino-acid linker in between the DBD and the CTD. To investigate further, we constructed a ParB (L224C I304C)-TEV variant, in which a TEV protease cleavage site was inserted within the DBD-CTD linker (*Figure 5—figure supplement 3A*). Again, ParB (L224C I304C)-TEV entrapped a circular *parS* plasmid efficiently in the presence of CTP (the HMW smear on lane 7, *Figure 5—figure supplement 3A*). However, a post-crosslinking treatment with TEV protease eliminated such HMW smear, presumably by creating a break in the polypeptide through which a circular plasmid could escape (lane 8, *Figure 5—figure supplement 3A*). We also extracted crosslinked ParB (L224C I304C) from gel slices that encompassed the HMW smear and electrophoresed the eluted proteins again on a denaturing gel to find a single band that migrated similarly to a double-crosslinked protein (lane 9, *Figure 5—figure supplement 2B*). Therefore, our results suggest that a ParB dimer, rather than ParB oligomers, is the major species that entraps DNA. Taken together, we suggest that *C. crescentus* ParB dimer functions as a molecular clamp that entraps *parS*-containing DNA within a DBD-CTD compartment upon CTP binding. This is also consistent with experiments that showed a premature and irreversible closing of ParB clamps, achieved either by an extended preincubation with CTPγS (*Jalal et al., 2020c* and *Figure 5—figure supplement 4B*) or by pre-crosslinking a closed clamp form of ParB (*Figure 5—figure supplement 4C*), prevented nucleation at *parS* and DNA entrapment.

## *C. crescentus* ParB (E102A) is a clamp-locked mutant that is defective in clamp reopening

Next, we investigated the potential role(s) of CTP hydrolysis. Hydrolysis is unlikely to be required for DNA entrapment and translocation since ParB in complex with CTPγS can still self-load and slide on DNA (*Jalal et al., 2020c*; *Soh et al., 2019*). *M. xanthus* ParB (N172A) and *B. subtilis* ParB (N112S) mutants, which bind but cannot hydrolyze CTP, failed to form higher-order protein-DNA complexes inside the cells (*Osorio-Valeriano et al., 2019*; *Soh et al., 2019*). However, these ParB variants are already impaired in NTD self-dimerization (*Soh et al., 2019*), hence the mechanistic role of CTP hydrolysis is still unclear. We postulated that creation of a ParB variant defective in CTP hydrolysis but otherwise competent in NTD self-dimerization would enable us to investigate the possible role of CTP hydrolysis. To this end, we performed alanine scanning mutagenesis on the CTP-binding pocket of *C. crescentus* ParB (*Figure 2C*). Eleven purified ParB variants were assayed for CTP binding by a membrane-spotting assay (DRaCALA) (*Figure 6A*), and for CTP hydrolysis by measuring the release rate of inorganic phosphate (*Figure 6B*). Moreover, their propensity for NTD self-dimerization was also analyzed by crosslinking with BMOE (*Figure 6C* and *Figure 6—figure supplement 1*). Lastly, their ability to nucleate, slide, and entrap a closed *parS* DNA substrate was investigated by a biolayer interferometry (BLI) assay (*Figure 6D* and *Figure 6—figure supplement 2A*). Immobilizing a dual biotin-labeled DNA on a streptavidin-coated BLI surface created a closed DNA substrate that can be entrapped by ParB-CTP clamps (*Figure 5—figure supplement 4A*; *Jalal et al., 2020c*). The BLI assay monitors wavelength shifts resulting from changes in the optical thickness of the probe surface during the association/dissociation of ParB with a closed DNA substrate in real time (*Figure 6—figure supplement 2*).

Overall, we identified several distinct classes of ParB mutants:

1. Class I: ParB (R60A), (R103A), (R104A), (R139A), (N136A), (G79S), and (S74A) did not bind or bound radiolabeled CTP only weakly (*Figure 6A*), thus also showed weak to no CTP hydrolysis (*Figure 6B*) or clamp-closing activity (*Figure 6C,D*).
2. Class II: ParB (Q58A) and (E135A) that are competent in CTP-binding (*Figure 6A*), but defective in CTP hydrolysis (*Figure 6B*) and in entrapping a closed *parS* DNA substrate (*Figure 6D*). We noted that ParB (Q58A) and ParB (E135A) already had an elevated crosslinking efficiency even

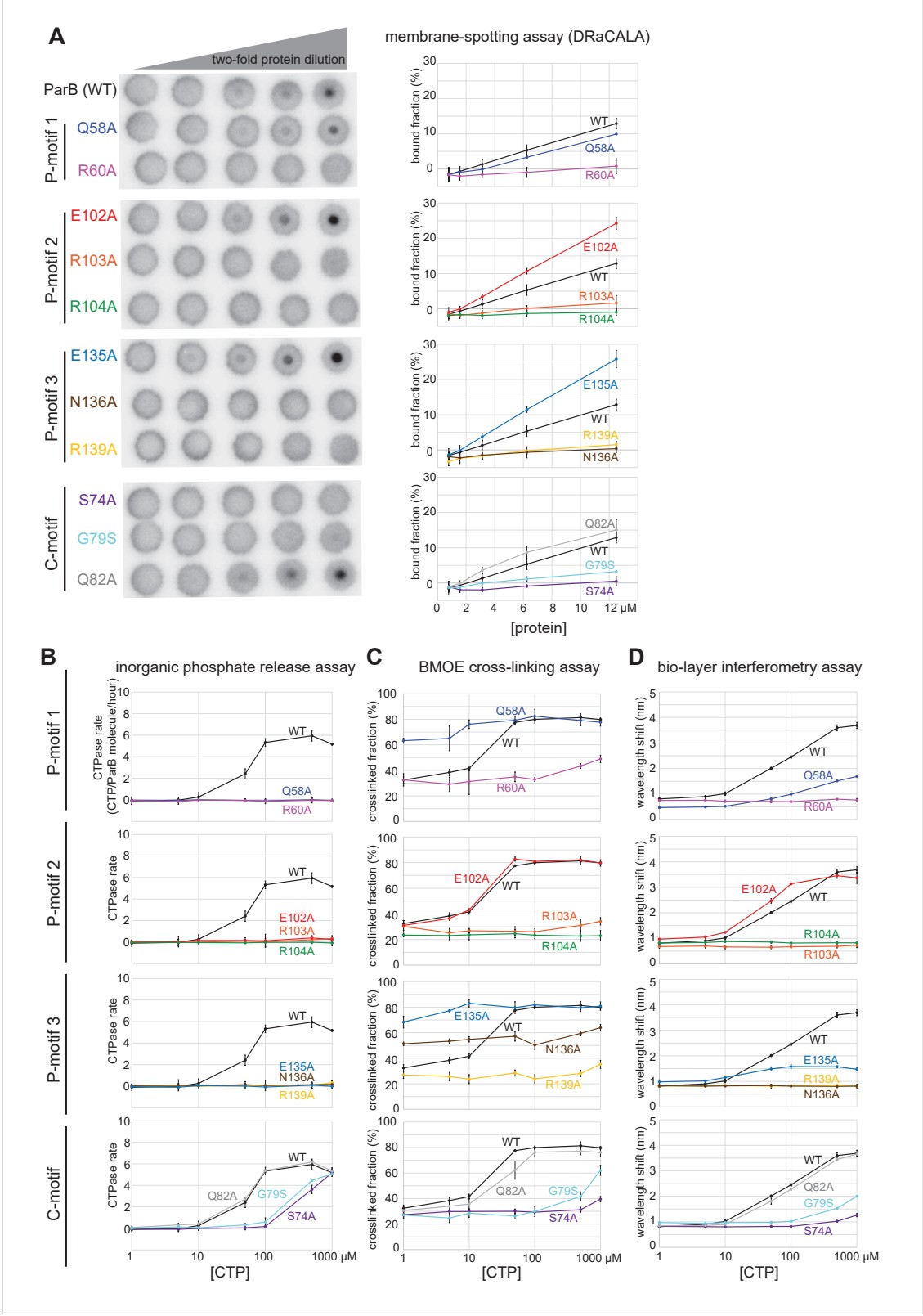

**Figure 6.** Alanine scanning mutagenesis of the *C. crescentus* ParB cytidine triphosphate (CTP)-binding pocket reveals several classes of clamp mutants. Eleven residues at C-motif and P-motifs 1–3 were individually substituted for alanine or glycine. (**A**) Membrane-spotting assay of ParB variants. CTP binding was monitored by membrane-spotting assay using radiolabeled CTP α-P[32]. The bulls-eye staining indicates CTP binding due to a more rapid immobilization of protein-ligand complexes compared to free ligands. All reactions contained various concentration of purified ParB, 5 nM radiolabeled

*Figure 6 continued on next page*

*Figure 6 continued*

CTP α-P³², 30 μM unlabeled CTP, and 1.5 μM 22 bp *parS* DNA. The bound fractions were quantified, and error bars represent SD from three replicates. All the reactions were spotted on the same membrane, the radiograph was rearranged solely for presentation purposes. (**B**) Inorganic phosphate release assay of ParB variants. The CTPase rates were measured at increasing concentration of CTP. All reactions contained 1 μM purified ParB variant, 0.5 μM 22 bp *parS* DNA, and an increasing concentration of CTP. (**C**) Bismaleimidoethane (BMOE) crosslinking assay of ParB variants. A second set of alanine scanning ParB variants, which harbor an additional Q35C substitution at the N-terminal domain (NTD), were also constructed and subsequently used in BMOE crosslinking experiments. Purified ParB variants (8 μM) were preincubated with 0.5 μM 22 bp *parS* DNA and an increasing concentration of CTP for 5 min before BMOE was added. Crosslinking products were resolved on a 12 % denaturing polyacrylamide gel and the crosslinked fractions were quantified (see also *Figure 6—figure supplement 1* for representation images). Error bars represent SD from three replicates. (**D**) Biolayer interferometry (BLI) assay of ParB variants. BLI analysis of the interaction between a premix of 1 μM ParB variant ± an increasing concentration of CTP and a 170 bp closed *parS* DNA substrate. See also *Figure 5—figure supplement 4A* for a schematic diagram of the BLI setup and *Figure 6—figure supplement 2* for representative BLI sensorgrams. BLI signal at the end of the association phase (± SD from three replicates) was plotted against CTP concentrations.

The online version of this article includes the following figure supplement(s) for figure 6:

**Source data 1.** Original files, annotation of the full raw gels, and data used to generate *Figure 6*.

**Figure supplement 1.** Denaturing polyacrylamide gel analysis of crosslinking products of alanine scanning ParB variants.

**Figure supplement 1—source data 1.** Original files, annotation of the full raw gels, and data used to generate *Figure 6—figure supplement 1*.

**Figure supplement 2.** Biolayer interferometry (BLI) analysis of the interaction between ParB variants and a 170 bp closed *parS* DNA substrate.

**Figure supplement 2—source data 1.** Data used to generate *Figure 6—figure supplement 2*.

in the absence of CTP (*Figure 6C*). This premature clamp closing might have resulted in a less than wild-type level of DNA entrapment (*Figure 6D*).

3. Class III: ParB (E102A) did not hydrolyze CTP (*Figure 6B*) but nevertheless bound CTP efficiently (*Figure 6A*) to self-dimerize at the NTD and to entrap DNA to the same level as ParB (WT) at all CTP concentrations (*Figure 6C,D*).

Upon a closer inspection of the BLI sensorgrams (*Figure 6—figure supplement 2B* and *Figure 7*), we noted that the entrapped ParB (E102A) did not noticeably dissociate from a closed DNA substrate when the probe was returned to a buffer-only solution (dissociation phase, $k_{off}$ = 8.0 × 10⁻⁴ ± 1.9 × 10⁻⁴ s⁻¹, *Figure 6—figure supplement 2B* and *Figure 7*). By contrast, entrapped ParB (WT) dissociated approximately 15-fold faster into buffer ($k_{off}$ = 1.2 × 10⁻² ± 3.7 × 10⁻⁴ s⁻¹). Further experiments showed that DNA entrapment by ParB (E102A), unlike ParB (WT), is more tolerant to high-salt solution (up to 1 M NaCl, *Figure 7A*). Nevertheless, ParB (E102A)-CTP could not accumulate on a BamHI-restricted open DNA substrate (*Figure 7B,C*; *Jalal et al., 2020c*), suggesting that ParB (E102A)-CTP, similar to ParB (WT), also form a closed clamp that runs off an open DNA end. Collectively, our results suggest that *parS* DNA and CTP induced a stably closed clamp conformation of ParB (E102A) in vitro.

To investigate the function of ParB (E102A) in vivo, we expressed a FLAG-tagged version of *parB* (E102A) from a vanillate-inducible promoter (P$_{van}$) in a *C. crescentus* strain where the native *parB* was under the control of a xylose-inducible promoter (P$_{xyl}$) (*Figure 8A*). Cells were depleted of the native ParB by adding glucose for 4 hr, subsequently vanillate was added for another hour before cells were fixed with formaldehyde for ChIP-seq. Consistent with the previous report (*Tran et al., 2018*), the ChIP-seq profile of FLAG-ParB (WT) showed an ~10 kb region of enrichment above background with clearly defined peaks that correspond to the positions of *parS* sites (*Figure 8A*). By contrast, the ChIP-seq profile of FLAG-ParB (E102A) is significantly reduced in height but has an extra peak over the *parB* coding sequence (*Figure 8A*, asterisk). The instability of FLAG-ParB (E102A) in its native *C. crescentus* host, and hence the reduced protein level (*Figure 8—figure supplement 1A*), might explain the overall lower height of its ChIP-seq profile (*Figure 8*). The reason for an extra peak over *parB* in the ChIP-seq profile of ParB (E102A) is still, however, unknown. We also noted that expressing ParB (E102A) could not rescue cells with depleted ParB (WT) (*Figure 8—figure supplement 2*). Again, due to the caveat of a lower ParB (E102A) protein level in *C. crescentus* (*Figure 8—figure supplement 1A*), we could not reliably link the in vitro properties of ParB (E102A) to its behaviors in the native host.

To overcome the caveat of protein instability, we instead investigated the spreading of ParB (WT) vs. ParB (E102A) from *parS* by analyzing the *C. crescentus* ParB/*parS* system in *Escherichia coli*. *E. coli* does not possess a ParA/ParB homolog nor a *parS*-like sequence, thus it serves as a suitable heterologous host. *C. crescentus parS* sites 3 and 4 were engineered onto the *E. coli* chromosome at the *ygcE* locus (*Figure 8B*). CFP-tagged ParB (WT/E102A) was expressed from a leaky lactose

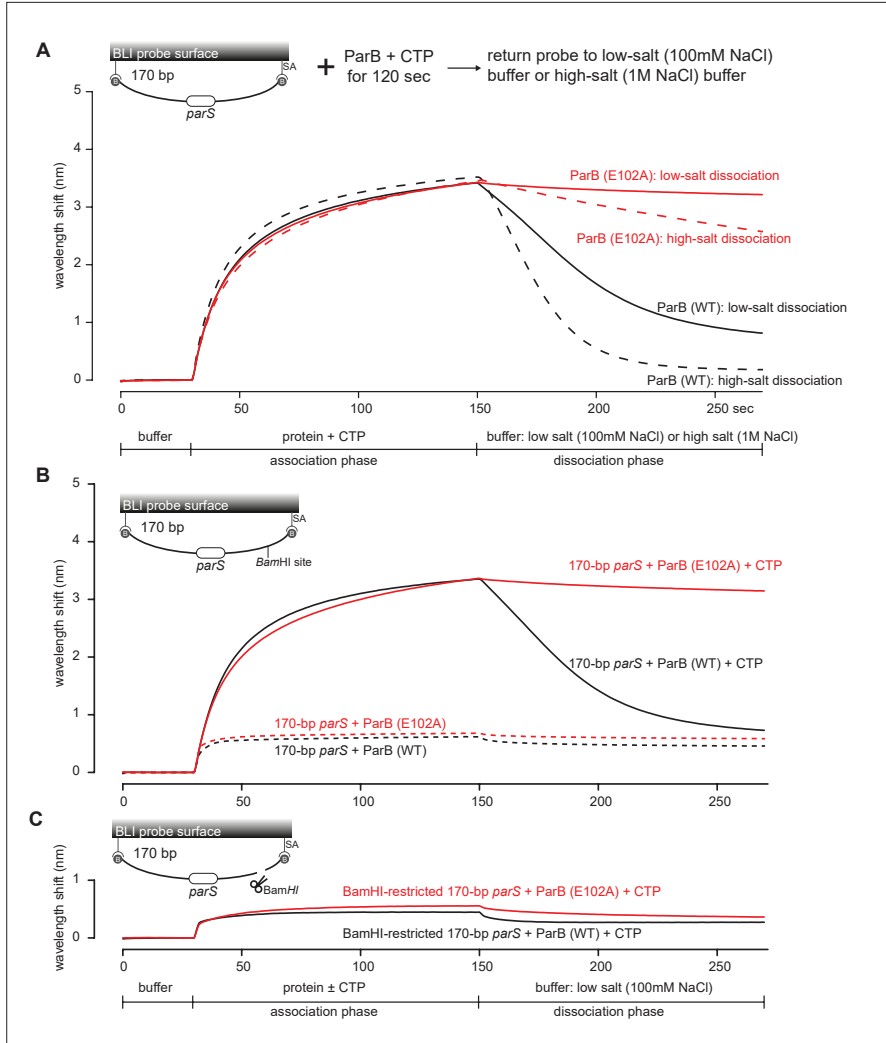

**Figure 7.** The DNA-entrapped ParB (E102A)-CTP clamp is resistant to high-salt conditions. (**A**) Biolayer interferometry (BLI) analysis of the interaction between a premix of 1 µM *C. crescentus* ParB (WT) or ParB (E102A) + 1 mM cytidine triphosphate (CTP) and 170 bp dual biotin-labeled *parS* DNA. For the dissociation phase, the probe was returned to a low-salt buffer that contains 100 mM NaCl (solid black or red lines) or to a high-salt buffer that contains 1 M NaCl (dashed black or red lines). The schematic diagram of the BLI probe shows a closed *parS* DNA substrate due to the interactions between a dual biotin-labeled DNA and the streptavidin (SA)-coated probe surface. (**B**) BLI analysis of the interaction between a premix of 1 µM *C. crescentus* ParB (WT) or ParB (E102A) + 1 mM CTP (solid lines) or –1 mM CTP (dashed lines) and 170 bp dual biotin-labeled *parS* DNA. (**C**) Same as panel (**B**) but immobilized DNA fragments have been restricted with BamHI before BLI analysis.

The online version of this article includes the following figure supplement(s) for figure 7:

**Source data 1.** Data used to generate *Figure 7*.

promoter (P*lac*, no IPTG was added) on a medium-copy-number plasmid. CFP-ParB (WT/E102A) was produced at the same level, as judged by an immunoblot (*Figure 8—figure supplement 1B*). We observed by ChIP-seq that CFP-ParB (WT) in an *E. coli* host spreads asymmetrically ~5 kb around *parS* sites. By contrast, the shape of the ParB (E102A) distribution was clearly different from that of ParB (WT); the profile was further expanded to both neighboring sides of *parS* (covering in total ~26 kb) at the expense of the enrichment at *parS* itself (*Figure 8B*). The more excessive spreading of ParB (E102A) might suggest that this variant, in the absence of CTP hydrolysis, persisted and perhaps slid further away from the loading site *parS* in *E. coli*. The reduced enrichment of ParB (E102A) at *parS* itself (*Figure 8B*) might be due to reduced cytoplasmic ParB (E102A) available to re-nucleate at *parS* and/or due to stably entrapped ParB (E102A) sterically hindering further nucleation events. We

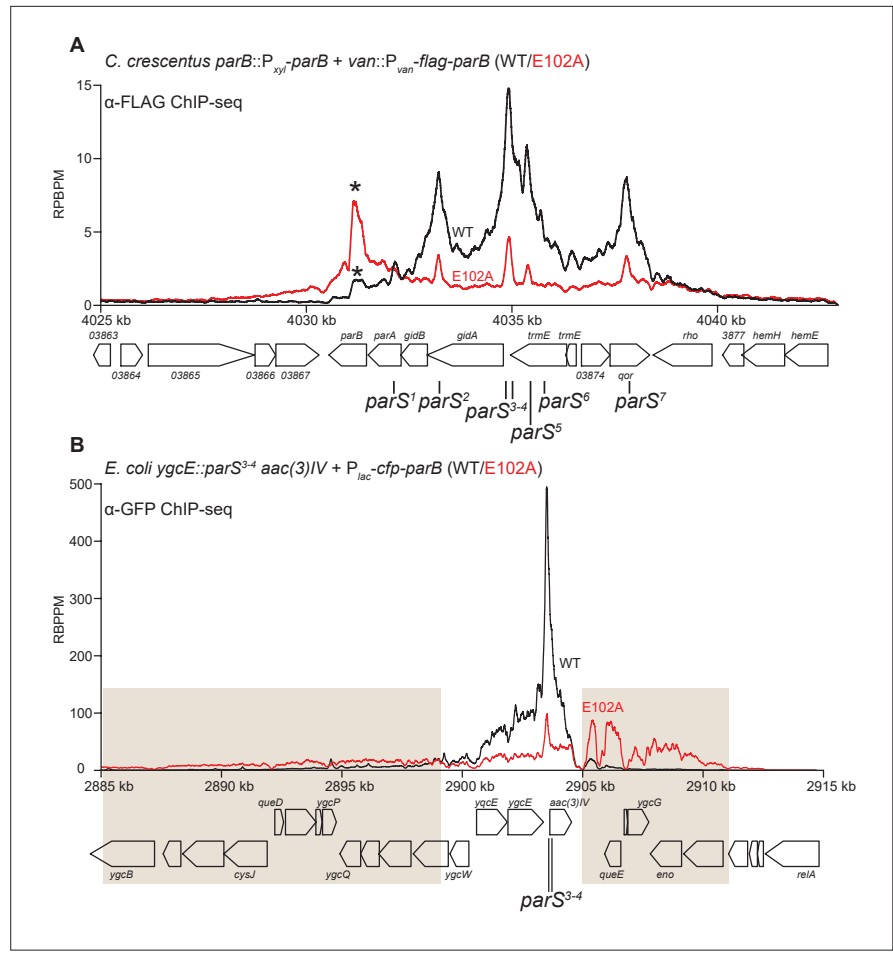

**Figure 8.** ParB (E102A) occupies a more extended DNA region surrounding *parS* sites than ParB (WT) in a heterologous host (*E. coli*) but not in the native host (*C. crescentus*). (**A**) ChIP-seq showed the distribution of FLAG-tagged ParB (WT) (black) and FLAG-ParB (E102A) (red) on *C. crescentus* chromosome between +4025 kb and +4042 kb. Underlying genes and *parS* sites are also shown below ChIP-seq profiles. An asterisk (*) indicates an extra peak over the *parB* coding sequence in the profile of FLAG-ParB (E102A). ChIP-seq signals were reported as the number of reads per base pair per million mapped reads (RPBPM). (**B**) ChIP-seq showed the distribution of CFP-tagged ParB (WT) (black) and CFP-ParB (E102A) (red) on an *E. coli* chromosome between +2885 kb and +2915 kb. *C. crescentus parS* sites 3 and 4 were engineered onto the *E. coli* chromosome at the *ygcE* locus. CFP-tagged ParB (WT/E102A) was expressed from a leaky lactose promoter (P$_{lac}$, no IPTG was added) on a medium-copy-number plasmid. Shaded boxes show areas with more enrichment in the ChIP-seq profile of CFP-ParB (E102A) compared to that of CFP-ParB (WT).

The online version of this article includes the following figure supplement(s) for figure 8:

**Figure supplement 1.** Immunoblot analysis of ParB (WT) vs. E102A.

**Figure supplement 1—source data 1.** Original files and annotation of the uncropped blots used to generate *Figure 8—figure supplement 1*.

**Figure supplement 2.** Expressing a FLAG-tagged version of ParB (E102A) could not complement the depletion of wild-type untagged ParB.

**Figure supplement 2—source data 1.** Original files, annotation of the full raw images, and data used to generate *Figure 8—figure supplement 2*.

**Figure supplement 3.** The fluorescence intensity of CFP-ParB (E102A) foci in an *E. coli* heterologous host is higher than that of CFP-ParB (WT).

**Figure supplement 3—source data 1.** Data used to generate *Figure 8—figure supplement 3*.

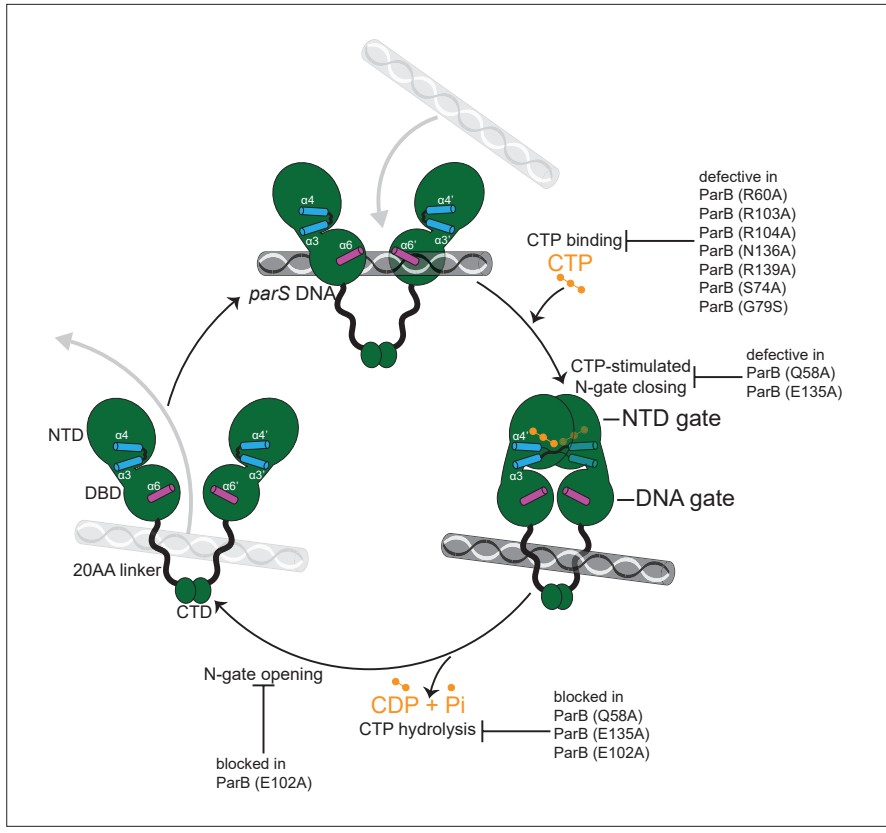

**Figure 9.** A model for *C. crescentus* ParB nucleating, sliding, and recycling cycle. ParB (dark green) consists of three domains: an N-terminal CTP-binding domain (NTD), a central *parS* DNA-binding domain (DBD), a C-terminal dimerization domain (CTD), and a 20 amino acid linker that connects the DBD and the CTD together. Nucleating ParB is an open clamp, in which *parS* DNA is captured at the DBD (the DNA-gate). Upon binding CTP (orange), the NTD self-dimerizes to close the NTD-gate of the clamp. CTP-binding and the exchange of helices α4 and α4' (blue) stabilize this closed conformation. The DBD also move closer together to close the DNA-gate, potentially driving *parS* DNA into a compartment between the DNA-gate and the C-terminal domain. In the nucleotide-bound state, the DBD and the DNA-recognition helices (α6 and α6', magenta) are incompatible with DNA binding. CTP hydrolysis and/or the release of hydrolytic products (CDP and inorganic phosphate Pi) may reopen the gates to release DNA. Substitutions that affect key steps in the CTP biding/hydrolysis cycle are also indicated on the schematic diagram.

also noted that the ChIP-seq profile of CFP-ParB (E102A) in *E. coli* is highly asymmetrical, with more enrichment in the 2905–2911 kb region than the 2885–2899 kb region (shaded areas, *Figure 8B*). The asymmetrical spreading is possibly due to an impediment in one direction by roadblocks such as RNA polymerases or DNA-bound proteins, which have been shown previously to be able to interfere with ParB spreading (*Balaguer F de et al., 2021*; *Breier and Grossman, 2007*; *Jalal et al., 2020c*; *Murray et al., 2006*; *Rodionov et al., 1999*; *Soh et al., 2019*).

Lastly, we quantified the fluorescence intensity of CFP-ParB (WT/E102A) foci inside cells and found a higher CFP signal for CFP-ParB (E102A) when compared with CFP-ParB (WT) (*Figure 8—figure supplement 3*). The higher intensity of the localizations could be due to more DNA-bound ParB (E102A) molecules surrounding the *parS* locus, which is consistent with the ChIP-seq observation showing CFP-ParB (E102A) occupying a more extended genomic area in *E. coli*. Altogether, at least in the heterologous *E. coli* host, the 'clamp-locked' phenotype of ParB (E102A) implies a possible role of CTP hydrolysis and/or the release of hydrolytic products in reopening wild-type ParB clamp to release DNA and to recycle ParB.

## Discussion

In this study, we provide structural insights into the nucleating and sliding states of *C. crescentus* ParB. Nucleating ParB is an open clamp in which *parS* DNA is held tightly (nM affinity) at the DBD

(*Tran et al., 2018*). The NTDs of nucleating ParB can adopt multiple alternative conformations, and crucially there is no contact between opposing NTDs. We liken this conformation of the NTD to that of an open gate (NTD-gate), through which *parS* DNA might gain access to the DBD (*Figure 9*). In the sliding state, CTP promotes the self-dimerization of the NTDs, thus closing the NTD-gate (*Figure 9*). Opposing DBDs also move approximately 10 Å closer together, bringing about a conformation that is DNA incompatible. Again, we liken this conformation of the DBDs to that of a closed gate (DNA-gate) (*Figure 9*). Overall, the DNA-gate closure explains how CTP binding might switch ParB from a nucleating to a sliding state.

Our data suggest that the closure of the two gates drives *parS* DNA into a compartment in between the DBD and the CTD. Previously, (*Soh et al., 2019*) compared the *B. subtilis* ParBΔCTD-CDP co-crystal structure to that of a *H. pylori* ParBΔCTD-*parS* complex and proposed that DNA must be entrapped in the DBD-CTD compartment (*Soh et al., 2019*). Here, the available structures of nucleating and sliding ParB from the same bacterial species enabled us to introduce a crosslinkable cysteine (L224C) at the DBD, and subsequently provided a direct evidence that the DBD-CTD compartment is the DNA-entrapping compartment. The linker that connects the DBD and the CTD together is not conserved in amino acid sequence among chromosomal ParB orthologs (*Figure 2—figure supplement 2*); however, we noted that the linker is invariably ~20 amino acid in length and positively charged lysines are over-represented (*Figure 2—figure supplement 2*). The biological significance of the linker length and its lysines, if any, is currently unknown. However, it is worth noting that a human PCNA clamp was proposed to recognize DNA via lysine-rich patches lining the clamp channel, and that these lysine residues help PCNA to slide by tracking the DNA backbone (*De March et al., 2017*). Investigating whether lysine residues in the DBD-CTD linker of ParB have a similar role is an important subject for the future.

If not already bound on DNA, the closed ParB clamp presumably cannot self-load onto *parS* owing to its inaccessible DBD. In this study, we showed that *parS* DNA promotes the CTP-dependent NTD-gate closure (*Figure 5B*), thus is likely a built-in mechanism to ensure gate closure results in a productive DNA entrapment. However, the molecular basis for the *parS*-enhanced gate closure remains unclear due to the lack of a crystal structure of *C. crescentus* apo-ParB, despite our extensive efforts.

CTP functions as a molecular latch that stabilizes the closure of the NTD-gate of ParB. Here, we provide evidence that CTP hydrolysis might contribute to reopening the closed NTD-gate. A previous structure of a *B. subtilis* ParBΔCTD-CDP complex also has its NTD-gate closed (CTP was hydrolyzed to CDP during the crystallization) (*Soh et al., 2019*), hence it is likely that both CTP hydrolysis and the subsequent release of hydrolytic products are necessary to reopen the gates. However, ParB has a weak to negligible affinity for CDP, hence the CDP-bound ParB species might be short-lived in solution and might not play a significant biological role. Once the clamp is reopened, entrapped DNA might escape via the same route that it first enters. Other well-characterized DNA clamps, for example, type II topoisomerases open their CTD to release trapped DNA. However, the CTDs of ParB are stably dimerized independently of *parS* and CTP (*Figure 5B*), hence we speculate that the CTD of ParB is likely to be impassable to the entrapped DNA. The released ParB clamp might re-nucleate on *parS* and bind CTP to close the gate, hence restarting the nucleation and sliding cycle. Such a recycling mechanism might provide a biological advantage since a ParB clamp once closed could otherwise become stably trapped on DNA and thus eventually diffuse too far from the *parS* locus, as evidenced by the ChIP-seq profile of the E102A variant (expressed in *E. coli*) that is defective in CTP hydrolysis (*Figure 8B*). However, how CTP hydrolysis contributes to the assembly of the centromere in *C. crescentus* is still unclear due to the caveat that ParB (E102A) is unstable in the native host.

The CTP-bound structure of a *M. xanthus* ParB-like protein, PadC, was solved to a high resolution (1.7 Å); however, PadC does not possess noticeable CTPase activity (*Osorio-Valeriano et al., 2019*). A co-crystal structure of *B. subtilis* ParB with CDP was also solved to a high resolution (1.8 Å) but represents a post-hydrolysis state instead. Lastly, our CTPγS-bound *C. crescentus* ParB crystals diffracted to 2.7 Å, thus preventing water molecules, including a potential catalytic water, from being assigned with confidence. Therefore, the mechanism of CTP hydrolysis by a ParB CTPase remains unresolved. Nevertheless, based on our alanine scanning experiment (*Figure 8*), we speculate that Q58 (P-motif 1) and E102 (P-motif 2) might be involved in the catalytic mechanism of *C. crescentus* ParB. Supporting this view, we noted that an equivalent Q37 in *B. subtilis* ParB does not contact the hydrolytic product CDP, and this residue is not conserved in the catalytic-dead *M. xanthus* PadC (F308,

which does not contact CTP, occupies this position in PadC instead) (*Figure 2—figure supplement 3*). E102 is also not conserved in M. *xanthus* PadC (F348 occupies this equivalent position) (*Figure 2—figure supplement 3*). Given that ParB is the founding member of a new CTPase protein family (*Jalal et al., 2020c*; *Osorio-Valeriano et al., 2019*; *Soh et al., 2019*), further studies are needed to fully understand the molecular mechanism of CTP hydrolysis so that the knowledge gained might be generalized to other CTPases.

Recently, an F-plasmid ParB was shown to form biomolecular condensates in vivo that might bridge distal ParB_F dimers together (*Guilhas et al., 2020*; *Walter et al., 2020*). If and how CTP binding/hydrolysis and the flexibility of the NTD contribute to this process is unclear and will be an important challenge for future studies. It is equally important to better understand the in vivo interaction between ParB and ParA now that CTP is in the picture. Recent in vitro work with ParAB_F showed that two protomers of a single ParB_F dimer interact with a ParA_F dimer in the absence of CTP (*Taylor et al., 2021*). However, two ParBF protomers from two distinct dimers interact with a ParAF dimer in the presence of CTP and *parS* (*Taylor et al., 2021*). Which mode of action is dominant in vivo for a chromosomal ParAB*S* systems and whether interacting with ParA further facilitates CTP hydrolysis by ParB are still unknown. Future works will provide important insights to better understand the mechanism of ParA-directed DNA segregation.

# Materials and methods

**Key resources table**

| Reagent type (species) or resource | Designation | Source or reference | Identifiers | Additional information |
|---|---|---|---|---|
| Strain, strain background (*Escherichia coli*) | See *Supplementary file 1A* | This paper | | See *Supplementary file 1A* |
| Strain, strain background (*Caulobacter crescentus*) | See *Supplementary file 1A* | This paper | | See *Supplementary file 1A* |
| Recombinant DNA reagent | See *Supplementary file 1B* | This paper | | See *Supplementary file 1B* |
| Sequence-based reagent | See *Supplementary file 1B* | This paper | | See *Supplementary file 1B* |
| Antibody | Anti-GFP antibody (HRP) (Rabbit polyclonal) | Abcam | Cat# ab190584 | Western blot (1:5000) |
| Antibody | Anti-GFP Sepharose beads | Abcam | Cat# ab69314 | For ChIP-seq experiments |
| Antibody | Anti-FLAG antibody (HRP) (Mouse monoclonal) | Merck | Cat# A8592 | Western blot (1:5000) |
| Antibody | Anti-FLAG M2 affinity agarose beads | Merck | Cat# A2220 | For ChIP-seq experiments |
| Commercial assay or kit | Amersham Protran supported western blotting membranes, nitrocellulose | GE Healthcare | Cat# GE10600016 | Pore size 0.45 µm, for DRaCALA assay |
| Commercial assay or kit | EnzChek Phosphate Assay Kit | ThermoFisher | Cat# E6646 | |
| Commercial assay or kit | Gibson Assembly Master Mix | NEB | Cat# E2611S | |
| Commercial assay or kit | Gateway BP Clonase II enzyme mix | ThermoFisher | Cat# 11789020 | |
| Commercial assay or kit | Dip-and-Read Streptavidin biosensors SAX2 | Sartorius UK | Cat# 18-5019 | |
| Commercial assay or kit | HisTrap High Performance column | GE Healthcare | Cat# GE17524801 | |
| Commercial assay or kit | HisTrap Heparin High Performance column | GE Healthcare | Cat# GE17040601 | |

*Continued on next page*

*Continued*

| Reagent type (species) or resource | Designation | Source or reference | Identifiers | Additional information |
|---|---|---|---|---|
| Commercial assay or kit | HiLoad 16/600 Superdex 200 pg column | GE Healthcare | Cat# GE28989335 | |
| Commercial assay or kit | 0.5 mL Zeba spin desalting columns | ThermoFisher | Cat# 89,882 | 7 K Da molecular weight cutoff |
| Peptide, recombinant protein | *Bam*HI-HF | NEB | Cat# R3136S | 20,000 units/mL |
| Peptide, recombinant protein | *Hind*III-HF | NEB | Cat# R3104S | 20,000 units/mL |
| Chemical compound, drug | Benzonase nuclease | Merck | Cat# E1014 | 250 units/μL |
| Chemical compound, drug | CTP | ThermoFisher | Cat# R0451 | 100 mM solution |
| Chemical compound, drug | CTPγS | Jena Bioscience | | Custom synthesis (purity ≥96%) |
| Chemical compound, drug | $P^{32}$-α-CTP | Perkin Elmer | Cat# BLU008H250UC | 3,000 Ci/mmol, 10 mCi/mL, 250 μCi |
| Chemical compound, drug | Bismaleimidoethane (BMOE) | ThermoFisher | Cat# 22323 | Dissolved in DMSO |
| Chemical compound, drug | AcTEV protease | ThermoFisher | Cat# 12575015 | 10 units/μL |
| Software, algorithm | BLItz Pro | Molecular Devices | Cat# 50-0156 | Version 1.2 |
| Software, algorithm | AIMLESS | *Evans and Murshudov, 2013* | http://www.ccp4.ac.uk/ | Version 0.7.4 |
| Software, algorithm | BUCCANEER | *Cowtan, 2006* | http://www.ccp4.ac.uk/ | Version 1.6.10 |
| Software, algorithm | Coot | *Emsley and Cowtan, 2004* | http://www.ccp4.ac.uk/ | Version 0.9.5 |
| Software, algorithm | CHAINSAW | *Stein, 2008* | http://www.ccp4.ac.uk/ | Version 7.0.077 |
| Software, algorithm | DIALS | *Winter et al., 2018* | https://dials.github.io | Version 3.1.0 |
| Software, algorithm | Excel 2016 | Microsoft | RRID:SCR_016137 | Version 16.0 |
| Software, algorithm | GraphPad Prism 8 | GraphPad Software | RRID:SCR_002798 | Version 8 |
| Software, algorithm | ImageJ | NIH | https://imagej.net/ RRID:SCR_003070 | Version 1.50 |
| Software, algorithm | Image Studio Lite | LI-COR Biosciences | RRID:SCR_013715 | Version 5.2 |
| Software, algorithm | PISA | *Krissinel, 2015* | http://www.ccp4.uk/pisa/ | Version 2.1.1 |
| Software, algorithm | MolProbity | *Williams et al., 2018* | http://molprobity.biochem.duke.edu/ | Version 4.5 |
| Software, algorithm | PHASER | *McCoy et al., 2007* | https://www.phenix-online.org/ | Version 2.8.2 |
| Software, algorithm | PyMOL | The PyMOL Molecular Graphics System | https://pymol.org/2/ | Version 2.4.0 |
| Software, algorithm | R | R Foundation for Statistical Computing | https://www.r-project.org/ | Version 3.2.4 |
| Software, algorithm | REFMAC5 | *Murshudov et al., 1997* | http://www.ccp4.ac.uk/ | Version 5.8.0258 |
| Software, algorithm | SCULPTOR | *Bunkóczi and Read, 2011* | http://www.ccp4.ac.uk/ | Version 0.0.3 |
| Software, algorithm | XDS | *Kabsch, 2010* | https://xds.mr.mpg.de/ | Version Nov11-2017 |
| Software, algorithm | XIA2 | *Winter, 2009* | https://xia2.github.io/index.html | Version 0.3.7.0 |

## Strains, media, and growth conditions

*E. coli* and *C. crescentus* were grown in LB and PYE, respectively. When appropriate, media were supplemented with antibiotics at the following concentrations (liquid/solid media for *C. crescentus*; liquid/solid media for *E. coli* [μg/mL]): carbenicillin (*E. coli* only: 50/100), chloramphenicol (1/2; 20/30), kanamycin (5/25; 30/50), and oxytetracycline (1/2; 12/12).

## Plasmids and strains construction

### Construction of pET21b::*parBΔCTD-(his)*6

The coding sequence of a C-terminally truncated *C. crescentus* ParB (ParBΔCTD, lacking the last 50 amino acids) was amplified by PCR using primers NdeI-Ct-ParB-F and HindIII-Ct-ParB-R, and pET21b::*parB-(his)₆* (*Lim et al., 2014*) as template. The pET21b plasmid backbone was generated via a double digestion of pET21b::*parB-(his)₆* with NdeI and HindIII. The resulting backbone was subsequently gel-purified and assembled with the PCR-amplified fragment of *parBΔCTD* using a 2 X Gibson master mix (NEB). Gibson assembly was possible owing to a 23 bp sequence shared between the NdeI-HindIII-cut pET21b backbone and the PCR fragment. These 23 bp regions were incorporated during the synthesis of primers NdeI-Ct-ParB-F and HindIII-Ct-ParB-R. The resulting plasmids were sequence verified by Sanger sequencing (Eurofins, Germany).

### Construction of pET21b::*parB-(his)*6 (WT and mutants)

DNA fragments containing mutated *parB* genes (*parB**) were chemically synthesized (gBlocks, IDT). The NdeI-HindIII-cut pET21b plasmid backbone and *parB** gBlocks fragments were assembled together using a 2 X Gibson master mix (NEB). Gibson assembly was possible owing to a 23 bp sequence shared between the NdeI-HindIII-cut pET21b backbone and the gBlocks fragment. The resulting plasmids were sequenced verified by Sanger sequencing (Genewiz, UK).

### pENTR::*attL1-parB* (WT/mutant)-*attL2*

The coding sequences of *C. crescentus* ParB (WT/mutants) were amplified by PCR and Gibson assembled into plasmid pENTR (Invitrogen) so that *parB* is flanked by phage attachment sites *attL1* and *attL2,* that is, Gateway cloning compatible. Correct mutations were verified by Sanger sequencing (Genewiz, UK).

### pMT571-1xFLAG-DEST

Plasmid pMT571 (*Thanbichler et al., 2007*) was first digested with NdeI and NheI. The plasmid backbone was gel-purified and eluted in 50 μL of water. The FLAG-*attR1-ccdB*-chloramphenicol^R-*attR2* cassette was amplified by PCR using primers P1952 and P1953, and pML477 as template. The resulting PCR fragment and the NdeI-NheI-cut pMT571 were assembled together using a 2 X Gibson master mix (NEB). Gibson assembly was possible owing to a 23 bp sequence shared between the two DNA fragments. These 23 bp regions were incorporated during the primer design to amplify the FLAG-*attR1-ccdB*-chloramphenicol^R-*attR2* cassette. The resulting plasmid was sequence verified by Sanger sequencing (Eurofins, Germany).

### pMT571-1xFLAG::*parB* (WT/mutants)

The *parB* (WT/mutant) genes were recombined into a Gateway-compatible destination vector pMT571-1xFLAG-DEST via LR recombination reaction (Invitrogen). For LR recombination reactions: 1 μL of purified pENTR::*attL1-parB* (WT/mutant)-*attL2* was incubated with 1 μL of the destination vector pMT571-1xFLAG-DEST, 1 μL of LR Clonase II master mix, and 2 μL of water in a total volume of 5 μL. The reaction was incubated for an hour at room temperature before being introduced into *E. coli* DH5α cells by heat-shock transformation. Cells were then plated out on LB agar+ tetracycline. Resulting colonies were restruck onto LB agar+ kanamycin and LB agar+ tetracycline. Only colonies that survived on LB+ tetracycline plates were subsequently used for culturing and plasmid extraction.

### pKTN25::*cfp-parB* (WT/E102A)

The coding sequence of ParB (WT/E102A) was amplified by PCR using primers P3392 and P3393, and pET21b::*C. crescentus* ParB (WT/E102A)-His₆ as template. The resulting DNA was gel-purified

and assembled with a BglII-EcoRI-cut pVCFPN-5 (*Thanbichler et al., 2007*) using a 2 X Gibson master mix, to result in vectors where the *cfp* is fused to the 5′-end of *parB* (WT/E102A). Gibson assembly was possible owing to a 23 bp sequence shared between the BglII-EcoRI-cut pVCFPN-5 backbone and the PCR fragment. To create vectors for expressing ParB (WT/E102A) in *E. coli*, the *cfp-parB* (WT/ E102A) segment was amplified by PCR using primers P3396 and P3397, and pVCFPN-5::*parB* (WT/ E102A) as template. The resulting DNA was then assembled with a HindIII-ClaI-cut pKTN25 (*Karimova et al., 1998*) using a 2 X Gibson master mix. Gibson assembly was possible owing to a 23 bp sequence shared between the HindIII-ClaI-cut pKTN25 backbone and the PCR fragment. Note that the double digestion with HindIII and ClaI removed the T25-encoding gene from the pKTN25 plasmid. The resulting vectors pKTN25::*cfp-parB* (WT/E102A) allow for the expression of CFP-tagged ParB (WT/E102A) from an IPTG-inducible lactose promoter ($P_{lac}$).

### Strains TLE1146 (AB1157 *ygcE*::260 bp *parS*::apramycin[R])

Lambda Red recombineering (*Datsenko and Wanner, 2000*) was used to insert a cassette consisting of 260 bp *C. crescentus* *parS*[3-4] sites and an apramycin antibiotic resistance gene *aac(3)IV* at the *ygcE* locus on the *E. coli* chromosome. To generate the first half of the cassette, DNA containing *parS*[3-4] sites was amplified by PCR using P1304 and P1305, and *C. crescentus* genomic DNA as template. To generate the second half of the cassette, DNA containing *aac(3)IV* was amplified by PCR using P1306 and P1307, and pIJ773 as template (*Gust et al., 2003*). The two resulting PCR products were gel-purified and joined together using a 2X Gibson master mix. The full-length 260 bp *parS*::apramycin[R] cassette was further amplified by PCR using P1304 and P1307. P1304 and P1307 also carry 49 bp homology to the left or the right of the insertion point at the *ygcE* locus. The resulting PCR product was gel-extracted and electroporated into an arabinose-induced *E. coli* AB1157/pKD46 cells. Colonies that formed on LB+ apramycin were restruck on LB+ apramycin and incubated at 42 °C to cure of pKD46 plasmid. Finally, the correct insertion of the *parS*-apramycin[R] cassette was verified by PCR and Sanger sequencing.

### Strains AB1157 + pKTN25::*cfp-parB* (WT/E102A)

*E. coli* AB1157 cells were made competent chemically and were transformed with pKTN25-*cfp-parB* (WT/E102A) to result in strains TLE3077 and TLE3078, respectively.

### Strains TLE1146 + pKTN25::*cfp-parB* (WT/E102A)

*E. coli* TLE1146 cells were made competent chemically and were transformed with pKTN25-*cfp-parB* (WT/E102A) to result in strains TLE3079 and TLE3080, respectively.

### Strains MT148 + pMT571-1xFLAG::*parB* (WT/mutants)

Electro-competent *C. crescentus* CN15N cells were electroporated with pMT571-1xFLAG::ParB (WT/ mutants) plasmid to allow for a single integration at the *vanA* locus. The correct integration was verified by PCR, and ΦCr30 phage lysate was prepared from this strain. Subsequently, *van*::$P_{van}$-*1xflag-parB* (WT/mutant), marked by a tetracycline[R] cassette, was transduced by phage ΦCr30 into MT148 (*Thanbichler and Shapiro, 2006*) to result in strains TLS3050-TLS3060.

## Protein overexpression and purification

Plasmid pET21b::*parBΔCTD-(his)₆* was introduced into *E. coli* Rosetta (DE3)-competent cells (Merck) by heat-shock transformation. 40 mL overnight culture was used to inoculate 4 L of LB medium + carbenicillin + chloramphenicol. Cells were grown at 37 °C with shaking at 250 rpm to an $OD_{600}$ of ~0.4. The culture was then left in the cold room to cool to 28 °C before isopropyl-β-D-thiogalactopyranoside (IPTG) was added at a final concentration of 0.5 mM. The culture was shaken for an additional 3 hr at 30 °C before cells were pelleted by centrifugation. Pelleted cells were resuspended in a buffer containing 100 mM Tris-HCl pH 8.0, 300 mM NaCl, 10 mM imidazole, 5 % (v/v) glycerol, 1 μL of Benzonase nuclease (Merck), 5 mg of lysozyme (Merck), and an EDTA-free protease inhibitor tablet (Merck). Cells were further lyzed by sonication (10 cycles of 15 s with 10 s resting on ice in between each cycle). The cell debris was removed through centrifugation at 28,000 g for 30 min and the supernatant was filtered through a 0.45 μm sterile filter (Sartorius). The protein was then loaded into a 1 mL HisTrap column (GE Healthcare) that had been pre-equilibrated with buffer A (100 mM Tris-HCl pH

8.0, 300 mM NaCl, 10 mM imidazole, and 5% [v/v] glycerol). Protein was eluted from the column using an increasing (10–500 mM) imidazole gradient in the same buffer. ParBΔCTD-containing fractions were pooled and diluted to a conductivity of 16 mS/cm before being loaded onto a 1 mL Heparin HP column (GE Healthcare) that had been pre-equilibrated with 100 mM Tris-HCl pH 8.0, 25 mM NaCl, and 5 % (v/v) glycerol. Protein was eluted from the Heparin column using an increasing (25 mM to 1 M NaCl) salt gradient in the same buffer. ParBΔCTD fractions were pooled and analyzed for purity by SDS-PAGE. Glycerol was then added to ParBΔCTD fractions to a final volume of 10 % (v/v), followed by 10 mM EDTA and 1 mM DTT. The purified ParBΔCTD was subsequently aliquoted, snap frozen in liquid nitrogen, and stored at –80 °C. ParBΔCTD that was used for X-ray crystallography was further polished via a gel-filtration column. To do so, purified ParBΔCTD was concentrated by centrifugation in an Amicon Ultra-15 3 kDa cutoff spin filters (Merck) before being loaded into a Superdex-200 gel filtration column (GE Healthcare). The gel filtration column was pre-equilibrated with buffer containing 10 mM Tris-HCl pH 8.0 and 250 mM NaCl. ParBΔCTD fractions were then pooled and analyzed for purity by SDS-PAGE.

Other C-terminally His-tagged ParB mutants were purified using HIS-Select Cobalt gravity flow columns as described previously (*Jalal et al., 2020b*). Purified proteins were desalted using a PD-10 column (Merck), concentrated using an Amicon Ultra-4 10 kDa cutoff spin column (Merck), and stored at –80 °C in a storage buffer (100 mM Tris-HCl pH 8.0, 300 mM NaCl, and 10 % [v/v] glycerol). Purified ParB mutants that were used in BMOE crosslinking experiments were buffer-exchanged and stored in a storage buffer supplemented with TCEP instead (100 mM Tris-HCl pH 7.4, 300 mM NaCl, 10 % [v/v] glycerol, and 1 mM TCEP).

Different batches of proteins were purified by ASBJ and NTT. Both biological (new sample preparations from a stock aliquot) and technical (same sample preparation) replicates were performed for assays in this study.

## DNA preparation for crystallization, EnzChek phosphate release assay, and differential radical capillary action of ligand assay (DRaCALA)

A 22 bp palindromic single-stranded DNA fragment (*parS*: GGATGTTTCACGTGAAACA TCC) (100 µM in 1 mM Tris-HCl pH 8.0, 5 mM NaCl buffer) was heated at 98 °C for 5 min before being left to cool down to room temperature overnight to form 50 µM double-stranded *parS* DNA. The core sequence of *parS* is underlined.

## Protein crystallization, structure determination, and refinement

Crystallization screens for the *C. crescentus* ParBΔCTD-*parS* complex were set up in sitting-drop vapor diffusion format in MRC2 96-well crystallization plates with drops comprising 0.3 µL precipitant solution and 0.3 µL of protein-DNA complex, and incubated at 293 K. His-tagged ParBΔCTD (~10 mg/mL) was mixed with a 22 bp *parS* duplex DNA at a molar ratio of 2:1.2 (protein monomer:DNA) in buffer containing 10 mM Tris-HCl pH 8.0 and 250 mM NaCl. The ParBΔCTD-*parS* crystals grew in a solution containing 20.5 % (w/v) PEG 3350, 260 mM magnesium formate, and 10 % (v/v) glycerol. After optimization of an initial hit, suitable crystals were cryoprotected with 20 % (v/v) glycerol and mounted in Litholoops (Molecular Dimensions) before flash-cooling by plunging into liquid nitrogen. X-ray data were recorded on beamline I04-1 at the Diamond Light Source (Oxfordshire, UK) using a Pilatus 6 M-F hybrid photon counting detector (Dectris), with crystals maintained at 100 K by a Cryojet cryocooler (Oxford Instruments). Diffraction data were integrated and scaled using XDS (*Kabsch, 2010*) via the XIA2 expert system (*Winter, 2009*) then merged using AIMLESS (*Evans and Murshudov, 2013*). Data collection statistics are summarized in *Table 1*. The majority of the downstream analysis was performed through the CCP4i2 graphical user interface (*Potterton et al., 2018*).

The ParBΔCTD-*parS* complex crystallized in space group $P2_1$ with cell parameters of *a* = 54.3, *b* = 172.9, *c* = 72.9 Å, and $\beta$ = 90.5° (*Table 1*). Analysis of the likely composition of the asymmetric unit (ASU) suggested that it contains four copies of the ParBΔCTD monomer and two copies of the 22 bp *parS* DNA duplex, giving an estimated solvent content of ~47 %.

Interrogation of the Protein Data Bank with the sequence of the *C. crescentus* ParBΔCTD revealed two suitable template structures for molecular replacement: apo-ParBΔCTD from *Thermus thermophilus* (*Leonard et al., 2004*) (PDB accession code: 1VZ0; 46 % identity over 82 % of the sequence) and *Helicobacter pylori* ParBΔCTD bound to *parS* DNA (*Chen et al., 2015*) (PDB accession code:

**Table 1.** X-ray data collection and processing statistics.

| Structure | *C. crescentus* ParBΔCTD-*parS* complex | *C. crescentus* ParBΔCTD CTPγS complex |
|---|---|---|
| *Data collection* | | |
| Diamond Light Source beamline | I04-1 | I03 |
| Wavelength (Å) | 0.916 | 0.976 |
| Detector | Pilatus 6 M-F | Eiger2 XE 16 M |
| Resolution range (Å) | 72.96–2.90 (3.08–2.90) | 70.59–2.73 (2.86–2.73) |
| Space group | $P2_1$ | $P2_1$ |
| Cell parameters (Å/°) | $a$ = 54.3, $b$ = 172.9, $c$ = 72.9, $\beta$ = 90.5 | $a$ = 69.5, $b$ = 56.1, $c$ = 71.4, $\beta$ = 98.4 |
| Total no. of measured intensities | 198,135 (33888) | 92,266 (8473) |
| Unique reflections | 29,654 (4775) | 14,516 (1756) |
| Multiplicity | 6.7 (7.1) | 6.4 (4.8) |
| Mean $I/\sigma(I)$ | 8.7 (1.4) | 5.4 (1.2) |
| Completeness (%) | 99.7 (100.0) | 98.8 (91.4) |
| $R_{merge}$* | 0.135 (1.526) | 0.195 (1.210) |
| $R_{meas}$† | 0.146 (1.646) | 0.212 (1.357) |
| $CC_{½}$‡ | 0.997 (0.677) | 0.991 (0.825) |
| Wilson $B$ value (Å$^2$) | 81.6 | 57.7 |
| *Refinement* | | |
| Resolution range (Å) | 72.96–2.90 (2.98–2.90) | 70.59–2.73 (2.80–2.73) |
| Reflections: working/free§ | 28155/1466 | 13824/678 |
| $R_{work}$¶ | 0.240 (0.366) | 0.248 (0.371) |
| $R_{free}$¶ | 0.263 (0.369) | 0.284 (0.405) |
| Ramachandran plot: favored/allowed/disallowed** (%) | 95.2/4.8/0 | 95.5/4.5/0 |
| R.m.s. bond distance deviation (Å) | 0.005 | 0.002 |
| R.m.s. bond angle deviation (°) | 1.05 | 1.19 |
| Mean $B$ factors: protein/DNA/other/ overall (Å$^2$) | 98/74/-/92 | 81/-/61/77 |
| PDB accession code | 6 T1F | 7BM8 |

Values in parentheses are for the outer resolution shell.

*$R_{merge} = \sum_{hkl} \sum_i |I_i(hkl)- \langle I(hkl)\rangle|/ \sum_{hkl} \sum_i I_i(hkl)$.

† $R_{meas} = \sum_{hkl} [N/(N-1)]^{1/2} \times \sum_i |I_i(hkl)- \langle I(hkl)\rangle|/ \sum_{hkl} \sum_i I_i(hkl)$, where $I_i(hkl)$ is the *i*th observation of reflection *hkl*, $\langle I(hkl)\rangle$ is the weighted average intensity for all observations *i* of reflection *hkl*, and *N* is the number of observations of reflection *hkl*.

‡$CC_{½}$ is the correlation coefficient between symmetry equivalent intensities from random halves of the dataset.

§The dataset was split into 'working' and 'free' sets consisting of 95% and 5% of the data, respectively. The free set was not used for refinement.

¶The R-factors $R_{work}$ and $R_{free}$ are calculated as follows: $R = \sum(| F_{obs} - F_{calc} |)/\sum| F_{obs} |$, where $F_{obs}$ and $F_{calc}$ are the observed and calculated structure factor amplitudes, respectively.

**As calculated using MolProbity (***Chen et al., 2010***).

4UMK; 42 % identity over 75 % of the sequence). First, single subunits taken from these two entries were trimmed using SCULPTOR (***Bunkóczi and Read, 2011***) to retain the parts of the structure that aligned with the *C. crescentus* ParBΔCTD sequence, and then all side chains were truncated to Cβ atoms using CHAINSAW (***Stein, 2008***). Comparison of these templates revealed a completely different relationship between the NTD and the DBD. Thus, we prepared search templates based on the individual domains rather than the subunits. The pairs of templates for each domain were then aligned and used as ensemble search models in PHASER (***McCoy et al., 2007***). For the DNA component, an ideal B-form DNA duplex was generated in COOT (***Emsley and Cowtan, 2004***) from

a 22 bp palindromic sequence of *parS*. A variety of protocols were attempted in PHASER (*McCoy et al., 2007*), the best result was obtained by searching for the two DNA duplexes first, followed by four copies of the DBD, giving a TFZ score of 10.5 at 4.5 Å resolution. We found that the placement of the DBDs with respect to the DNA duplexes was analogous to that seen in the *H. pylori* ParBΔCTD-*parS* complex. After several iterations of rebuilding in COOT and refining the model in REFMAC5 (*Murshudov et al., 1997*), it was possible to manually dock one copy of the NTD template (from 1VZ0) into weak and fragmented electron density such that it could be joined to one of the DBDs. A superposition of this more complete subunit onto the other three copies revealed that in only one of these did the NTD agree with the electron density. Inspection of the remaining unfilled electron density showed evidence for the last two missing NTDs, which were also added by manual docking of the domain template (from 1VZ0). For the final stages, TLS refinement was used with a single TLS domain defined for each protein chain and for each DNA strand. The statistics of the final refined model, including validation output from MolProbity (*Chen et al., 2010*), are summarized in *Table 1*.

Crystallization screens for the *C. crescentus* ParBΔCTD-CTPɣS complex crystal were also set up in sitting-drop vapor diffusion format in MRC2 96-well crystallization plates with drops comprising 0.3 μL precipitant solution and 0.3 μL of protein solution (~10 mg/mL) supplemented with 1 mM CTPɣS (Jena Biosciences) and 1 mM MgCl$_2$, and incubated at 293 K. The ParBΔCTD-CTPɣS crystals grew in a solution containing 15 % (w/v) PEG 3350, 0.26 M calcium acetate, 10 % (v/v) glycerol, 1 mM CTPɣS, and 1 mM MgCl$_2$. Suitable crystals were cryoprotected with 20 % (v/v) glycerol and mounted in Litholoops (Molecular Dimensions) before flash-cooling by plunging into liquid nitrogen. X-ray data were recorded on beamline I03 at the Diamond Light Source (Oxfordshire, UK) using an Eiger2 XE 16 M hybrid photon counting detector (Dectris), with crystals maintained at 100 K by a Cryojet cryocooler (Oxford Instruments). Diffraction data were integrated and scaled using DIALS (*Winter et al., 2018*) via the XIA2 expert system (*Winter, 2009*), then merged using AIMLESS (*Evans and Murshudov, 2013*). Data collection statistics are summarized in *Table 1*. The majority of the downstream analysis was performed through the CCP4i2 graphical user interface (*Potterton et al., 2018*).

The ParBΔCTD-CTPɣS complex crystallized in space group $P2_1$ with cell parameters of $a$ = 69.5, $b$ = 56.1, $c$ = 71.4 Å, and $\beta$ = 98.4° (*Table 1*). Analysis of the likely composition of the ASU suggested that it contains two copies of the ParBΔCTD monomer giving an estimated solvent content of ~50 %. Molecular replacement templates were generated from the ParBΔCTD-*parS* complex solved above. Attempts to solve the structure in PHASER using individual subunits taken from the latter in both conformations did not yield any convincing solutions, suggesting that the subunits had adopted new conformations. Given that the two subunit conformations observed in the previous structure differed largely in the relative dispositions of DBD and NTDs, we reasoned that a better outcome might be achieved by searching for the DBD and NTD separately. This time PHASER successfully placed two copies of each domain in the ASU such that they could be reconnected to give two subunits in a new conformation. The result was subjected to 100 cycles of jelly-body refinement in REFMAC5 before rebuilding with BUCCANEER (*Cowtan, 2006*) to give a model in which 77 % of the expected residues had been fitted into two chains and sequenced. The model was completed after further iterations of model editing in COOT and refinement with REFMAC5. In this case, TLS refinement was not used as this gave poorer validation results. The statistics of the final refined model, including validation output from MolProbity (*Chen et al., 2010*), are summarized in *Table 1*.

## Measurement of protein-DNA interaction by BLI assay

BLI experiments were conducted using a BLItz system equipped with High Precision Streptavidin 2.0 (SAX2) Biosensors (Molecular Devices). BLItz monitors wavelength shifts (nm) resulting from changes in the optical thickness of the sensor surface during association or dissociation of the analyte. All BLI experiments were performed at 22 °C. The streptavidin biosensor was hydrated in a low-salt-binding buffer (100 mM Tris-HCl pH 8.0, 100 mM NaCl, 1 mM MgCl$_2$, and 0.005 % [v/v] Tween 20) for at least 10 min before each experiment. Biotinylated double-stranded DNA (dsDNA) was immobilized onto the surface of the SA biosensor through a cycle of baseline (30 s), association (120 s), and dissociation (120 s). Briefly, the tip of the biosensor was dipped into a binding buffer for 30 s to establish the baseline, then to 1 μM biotinylated dsDNA for 120 s, and finally to a low-salt-binding buffer for 120 s to allow for dissociation.

After the immobilization of DNA on the sensor, association reactions were monitored at 1 µM dimer concentration of ParB with an increasing concentration of CTP (0, 1, 5, 10, 50, 100, 500, 1000 µM) for 120 s. At the end of each binding step, the sensor was transferred into a protein-free binding buffer to follow the dissociation kinetics for 120 s. The sensor can be recycled by dipping in a high-salt buffer (100 mM Tris-HCl pH 8.0, 1000 mM NaCl, 10 mM EDTA, and 0.005 % [v/v] Tween 20) for 5 min to remove bound ParB.

For the dissociation step in the BLI experiments in *Figure 7A*, the probe was returned to either a low-salt-binding buffer (100 mM Tris-HCl pH 8.0, 100 mM NaCl, 1 mM $MgCl_2$, and 0.005 % [v/v] Tween 20) for 30 s or a high-salt buffer (100 mM Tris-HCl pH 8.0, 1 M NaCl, 1 mM $MgCl_2$, and 0.005 % [v/v] Tween 20) for 30 s.

For experiments in *Figure 7C*, DNA-coated tips were dipped into 300 µL of restriction solution (266 µL of water, 30 µL of 10× buffer CutSmart [NEB], and 3 µL of BamHI-HF restriction enzyme [20,000 units/mL]) for 2 hr at 37 °C. As a result, closed DNA on the BLI surface was cleaved to generate a free DNA end.

For experiments in *Figure 5—figure supplement 4C*, purified ParB (Q35C) was incubated with 1 mM CTPγS in a binding buffer (100 mM Tris-HCl pH 7.4, 100 mM NaCl, 1 mM $MgCl_2$) for 30 min before BMOE was added to 1 mM. DTT was then added to the final concentration of 1 mM to quench the reaction. Subsequently, crosslinked ParB (Q35C) was buffer-exchanged into a storage buffer (100 mM Tris-HCl pH 8.0, 300 mM NaCl, and 10 % glycerol) using 0.5 mL Zeba desalting columns (ThermoFisher). BLITZ assays were performed using 5 µM dimer concentration of crosslinked ParB (Q35C) ± 1 mM CTP.

All sensorgrams recorded during BLI experiments were analyzed using the BLItz analysis software (BLItz Pro version 1.2, Molecular Devices) and replotted in R for presentation. Each experiment was triplicated, standard deviations were calculated in Excel, and a representative sensorgram is presented in *Figure 5—figure supplement 4*, *Figure 6—figure supplement 2*, and *Figure 7*.

## Differential radical capillary action of ligand assay (DRaCALA) or membrane-spotting assay

Purified *C. crescentus* ParB-His₆ (WT and mutants, at final concentrations of 0.7, 1.5, 3.1, 6.2, and 12.5 µM) were incubated with 5 nM radiolabeled $P^{32}$-α-CTP (Perkin Elmer), 30 µM of unlabeled CTP (ThermoFisher), and 1.5 µM of 22 bp *parS* DNA duplex in the binding buffer (100 mM Tris pH 8.0, 100 mM NaCl, and 10 mM $CaCl_2$) for 10 min at room temperature. 4 µL of samples were spotted slowly onto a nitrocellulose membrane and air-dried. The nitrocellulose membrane was wrapped in cling film before being exposed to a phosphor screen (GE Healthcare) for 2 min. Each DRaCALA assay was triplicated, and a representative autoradiograph was shown. Data were quantified using Multi-Gauge software 3.0 (Fujifilm), the bound fraction were quantified as described previously (*Roelofs et al., 2011*). Error bars represent standard deviations from triplicated experiments.

## Measurement of CTPase activity by EnzChek phosphate release assay

CTP hydrolysis was monitored using an EnzCheck Phosphate Assay Kit (ThermoFisher). Samples (100 µL) containing a reaction buffer supplemented with an increasing concentration of CTP (0, 1, 5, 10, 50, 100, 500, and 1000 µM), 0.5 µM of 22 bp *parS* DNA, and 1 µM ParB (WT or mutants) were assayed in a Biotek EON plate reader at 25 °C for 8 hr with readings every minute. The reaction buffer (1 mL) typically contained 740 µL Ultrapure water, 50 µL 20× reaction buffer (100 mM Tris pH 8.0, 2 M NaCl, and 20 mM $MgCl_2$), 200 µL MESG substrate solution, and 10 µL purine nucleoside phosphory-lase enzyme (one unit). Reactions with buffer only or buffer + CTP + 22 bp *parS* DNA only were also included as controls. The plates were shaken at 280 rpm continuously for 8 hr at 25 °C. The inorganic phosphate standard curve was also constructed according to the manual. The results were analyzed using Excel and the CTPase rates were calculated using a linear regression fitting in Excel. Error bars represent standard deviations from triplicated experiments.

## In vitro crosslinking assay using a sulfhydryl-to-sulfhydryl crosslinker BMOE

A 50 µL mixture of 8 µM ParB mutants (with residues at specific positions in the NTD, DBD, or CTD substituted to cysteine) ± CTP (0–1000 µM) ± 0.5 µM DNA (a 22 bp linear DNA or a 3 kb circular *parS/*

scrambled *parS* plasmid) was assembled in a reaction buffer (10 mM Tris-HCl pH 7.4, 100 mM NaCl, and 1 mM MgCl$_2$) and incubated for 5 min at room temperature. BMOE (1 mM final concentration from a 20 mM stock solution) was then added, and the reaction was quickly mixed by three pulses of vortexing. SDS-PAGE sample buffer containing 23 mM β-mercaptoethanol was then added immediately to quench the crosslinking reaction. Samples were heated to 50 °C for 5 min before being loaded on 12 % Novex WedgeWell Tris-Glycine gels (ThermoFisher). Protein bands were stained with an InstantBlue Coomassie solution (Abcam) and band intensity was quantified using Image Studio Lite version 5.2 (LI-COR Biosciences). The crosslinked fractions were averaged, and their standard deviations from triplicated experiments were calculated in Excel.

For the experiment described in lane 8 of *Figure 5C,D* and *Figure 5—figure supplement 2*, crosslinking reactions were performed as described above; however, the reaction was quenched using a quenching buffer (10 mM Tris-HCl pH 7.4, 100 mM NaCl, 1 mM MgCl$_2$, and 2.3 mM β-mercaptoethanol) instead. Subsequently, 1 μL of a non-specific DNA nuclease (Benzonase, 250 units/μL, Merck) was added, and the mixture was incubated at room temperature for a further 10 min before SDS-PAGE sample buffer was added. Samples were heated to 50 °C for 5 min before being loaded on 4–12% Novex WedgeWell Tris-Glycine gels (ThermoFisher).

For the experiments described in lane 8 of *Figure 5—figure supplement 3A*, crosslinking and quenching reactions were performed as described above before 1 μL of TEV protease (10 units/μL, ThermoFisher) was added. The mixture was incubated at room temperature for a further 30 min before SDS-PAGE sample buffer was added. Samples were heated to 50 °C for 5 min before being loaded on 4–12% Novex WedgeWell Tris-Glycine gels.

For experiments described in lane 9 of *Figure 5—figure supplement 3B*, proteins were released from gel slices by a 'crush & soak' method. Briefly, 10 gel slices were cut out from unstained SDS-PAGE gels and transferred to a 2 mL Eppendorf tube. Gel slices were frozen in liquid nitrogen and were crushed using a plastic pestle. The resulting paste was soaked in 500 μL of soaking buffer (10 mM Tris-HCl pH 8, 100 mM NaCl, 1 mM MgCl$_2$, and 1 μL of Benzonase [250 units/μL]), and the tube was incubated with rotation in a rotating wheel overnight. On the next day, the tube was centrifuged at 13,000 rpm for 5 min and the supernatant was transferred to a new 1.5 mL Eppendorf tube. The sample volume was reduced to ~50 μL using a SpeedVac vacuum concentrator before SDS-PAGE sample buffer was added in. The entire sample was loaded onto a single well of a 4–12% WedgeWell Tris-Glycine gel.

For experiments described in *Figure 5—figure supplement 1*, a circular *parS*-harboring plasmid was linearized at an unique HindIII site by HindIII-HF restriction enzyme. After restriction, the linearized DNA was extracted with phenol-chloroform and ethanol precipitated before being used in double-crosslinking experiments.

Polyacrylamide gels were submerged in an InstantBlue Coomassie solution (Abcam) to stain for protein or in a SYBR Green solution (ThermoFisher) to stain for DNA. Denatured samples were also loaded on 1 % TAE agarose gels and electrophoresed at 120 V for 40 min at room temperature. Afterwards, agarose gels were submerged in a SYBR green solution to stain for DNA.

## Chromatin immunoprecipitation with deep sequencing (ChIP-Seq)

α-FLAG ChIP-seq experiments on formaldehyde-fixed *C. crescentus* cells, and the subsequent data analysis was performed exactly as reported previously (*Tran et al., 2018*).

For ChIP-seq experiments on fixed *E. coli* cells, cells harboring pKTN25-*cfp-parB* (WT) or pKTN25-*cfp-parB* (E102A) were grown in 50 mL LB at 30 °C to mid exponential phase (OD$_{600}$ ~ 0.4, no IPTG was added). Subsequently, formaldehyde is added to a final concentration of 1 % to fix the cells. All following steps are identical to ChIP-seq for *C. crescentus*, except that α-GFP antibody coupled to sepharose beads (Abcam) was used to immunoprecipitate CFP-tagged ParB–DNA complexes.

Each ChIP-seq experiment was duplicated using biological replicates. For a list of ChIP-seq experiments and their replicates in this study, see *Supplementary file 1C*.

## Immunoblot analysis

For western blot analysis, *C. crescentus* or *E. coli* cells were pelleted and resuspended directly in 1× SDS sample buffer, then heated to 95 °C for 5 min before loading. Total protein was run on 12 % Novex WedgeWell gels (ThermoFisher) at 150 V for separation. The same amount of total protein was

loaded on each lane. Resolved proteins were transferred to PVDF membranes using the Trans-Blot Turbo Transfer System (BioRad) and probed with either a 1:5000 dilution of α-FLAG HRP-conjugated antibody (Merck) antibody or a 1:5000 dilution of α-GFP HRP-conjugated antibody (Abcam). Blots were imaged after incubation with SuperSignal West PICO PLUS Chemiluminescent Substrate (ThermoFisher) using an Amersham Imager 600 (GE Healthcare). Western blot experiments were duplicated using biological replicates.

### Fluorescence microscopy and image analysis

TLS3079 and TLS3080 were grown in M9 media supplemented with kanamycin (30 µg/mL) until $OD_{600}$ ~ 0.1 prior to imaging. The expression of *cfp-parB (WT/E102A)* was induced with 0.25 mM IPTG in culture for 60 min before imaging. Imaging was performed using a wide-field epifluorescence microscope (Eclipse Ti-2E, Nikon) with a 63× oil immersion objective (NA 1.41), illumination from pE4000 light source, Hamamatsu Orca Flash 4.0 camera, and a motorized XY stage. Images were acquired using NIS-elements software (version 5.1). For imaging in the CFP channel, 435 nm excitation wavelength was used with 1 s exposure.

Images were analyzed using ImageJ, and plots were generated in GraphPad Prism 8.0. For extracting information on number of ParB foci per cell as well as intensity of ParB in foci, the following analysis pipeline was implemented: cell masks were generated in ImageJ using analyze particle function on thresholds applied to phase profiles. Separately, even background subtraction function was applied to fluorescence profiles, images were convolved (using 'subtract background' and 'convolve' functions in ImageJ), and regions of interest (ROIs) for foci were detected via an application of appropriate thresholds. The cell masks and ROIs thus detected were applied to the raw data (after background correction) to extract intensity information for each ROI as well as total cell fluorescence. ROI intensity was plotted as a ratio of intensity within a focus ($intensity_{loc}$) normalized to total cell intensity ($intensity_{total}$). Along with intensity measurement, number of foci per cell was also recorded. The pipeline was implemented in ImageJ using the following command:

n = roiManager("count");for (j = 0; j < n; j++){roiManager("Select", j);run("Analyze Particles...", "size = 3–10 circularity = 0.40–1.00 display summarize add");}

## Acknowledgements

This study was funded by the Royal Society University Research Fellowship (URF\R\201020), BBSRC grant (BB/P018165/1), and a Wellcome Trust grant (221776/Z/20/Z, to TBKL), and a DST-SERB CRG grant 2019/003321 (to AB). ASBJ's PhD studentship was funded by the Royal Society (RG150448), and NTT was funded by the BBSRC grant-in-add (BBS/E/J/000PR9791 to the John Innes Centre). We thank Diamond Light Source for access to beamlines I04-1 and I03 under proposals MX13467 and MX18565 with support from the European Community's Seventh Framework Program (FP7/2007-2013) under Grant Agreement 283570 (BioStruct-X). We thank Stephan Gruber, Martin Thanbichler, and Manuel Osorio-Valeriano for sharing unpublished results.

## Additional information

### Funding

| Funder | Grant reference number | Author |
|---|---|---|
| Royal Society | URF\R\201020 | Tung B K Le |
| Royal Society | RG150448 | Adam S B Jalal |
| Biotechnology and Biological Sciences Research Council | BB/P018165/1 | Tung B K Le |
| Biotechnology and Biological Sciences Research Council | BBS/E/J/000PR9791 | Ngat T Tran |
| Wellcome Trust | 221776/Z/20/Z | Tung B K Le |

| Funder | Grant reference number | Author |
|---|---|---|
| Science and Engineering Research Board | 2019/003321 | Anjana Badrinarayanan |

The funders had no role in study design, data collection and interpretation, or the decision to submit the work for publication.

## Author contributions

Adam SB Jalal, Formal analysis, Investigation, Methodology, Writing – original draft; Ngat T Tran, Formal analysis, Investigation; Clare EM Stevenson, Investigation; Afroze Chimthanawala, Formal analysis, Investigation, Writing – review and editing; Anjana Badrinarayanan, Formal analysis, Investigation, Supervision, Writing – review and editing; David M Lawson, Formal analysis, Investigation, Methodology, Supervision, Writing – original draft; Tung BK Le, Conceptualization, Formal analysis, Funding acquisition, Investigation, Methodology, Supervision, Visualization, Writing – original draft, Writing – review and editing

## Author ORCIDs

Adam SB Jalal http://orcid.org/0000-0001-7794-8834
Ngat T Tran http://orcid.org/0000-0002-7186-3976
Anjana Badrinarayanan http://orcid.org/0000-0001-5520-2134
David M Lawson http://orcid.org/0000-0002-7637-4303
Tung BK Le http://orcid.org/0000-0003-4764-8851

## Decision letter and Author response

Decision letter https://doi.org/10.7554/eLife.69676.sa1
Author response https://doi.org/10.7554/eLife.69676.sa2

# Additional files

## Supplementary files

• Supplementary file 1. (A) Bacterial strains used in this study. (B) Plasmids, DNA, and protein sequences used in this study. (C) ChIP-seq datasets generated in this study.

• Transparent reporting form

## Data availability

The accession number for the sequencing data reported in this paper is GSE168968. Atomic coordinates for protein crystal structures reported in this paper were deposited in the RCSB Protein Data Bank with the accession number 6T1F and 7BM8. All of these sequencing and X-ray crystallography data are already open to the public. All other data generated or analyzed during this study are included in the manuscript.

The following dataset was generated:

| Author(s) | Year | Dataset title | Dataset URL | Database and Identifier |
|---|---|---|---|---|
| Le TBK | 2021 | A CTP-dependent gating mechanism enables ParB spreading on DNA in Caulobacter crescentus | https://www.ncbi.nlm.nih.gov/geo/query/acc.cgi?acc=GSE168968 | NCBI Gene Expression Omnibus, GSE168968 |
| Jalal ASB, Pastrana CL, Tran NT, Stevenson CEM, Lawson DM, Moreno-Herrero F, Le TBK | 2020 | Crystal structure of the C-terminally truncated chromosome-partitioning protein ParB from Caulobacter crescentus complexed to the centromeric parS site | https://www.rcsb.org/structure/6T1F | RCSB Protein Data Bank, 6T1F |

*Continued on next page*

*Continued*

| Author(s) | Year | Dataset title | Dataset URL | Database and Identifier |
|---|---|---|---|---|
| Jalal ASB, Pastrana CL, Tran NT, Stevenson CEM, Lawson DM, Moreno-Herrero F, Le TBK | 2021 | Crystal structure of the C-terminally truncated chromosome-partitioning protein ParB from Caulobacter crescentus complexed with CTP-gamma-S | https://www.rcsb.org/structure/7BM8 | RCSB Protein Data Bank, 7BM8 |

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
