## [Decision Letter]

**Acceptance summary:**

Bacterial ParB partition proteins have the novel property that they employ an unusual nucleotide cofactor for complex assembly at their specific DNA binding site, parS. The impact of this study is on our general understanding of this novel class of nucleotide-dependent processes, and the role that nucleotide-protein interactions play in DNA binding and bacterial physiology.

**Decision letter after peer review:**

Thank you for submitting your article "A CTP-dependent gating mechanism enables ParB spreading on DNA" for consideration by *eLife*. Your article has been reviewed by 3 peer reviewers, including Anthony G Vecchiarelli as the Reviewing Editor and Reviewer #1, and the evaluation has been overseen by Gisela Storz as the Senior Editor.

Essential revisions:

All three reviewers agree that the work is exciting, high quality, will be of broad interest, and should be published in *eLife*. However, in the "Recommendations to the Authors" there are two control experiments that all three reviewers agree are simple to perform and required. Also, all three reviewers had concerns regarding the ChIP-seq data that need to be addressed prior to publication.

1) From Reviewer 1: A control experiment where the authors pre-treat ParB with cross-linker *before* addition of DNA substrate to show that premature and irreversible closing of the ring prevents interactions with parS as well as spreading/sliding. The pre-crosslinking might alter protein conformation in other ways (besides closing a gate) that might destroy the DNA binding site (so there are caveats). Nevertheless if crosslinking did not prevent DNA binding, that would affect the overall story.

2) From Reviewer 2: A linearized plasmid control. The authors use the binding to circular DNA as evidence that ParB is topologically constrained, compared to a 22 bp linear DNA substrate. But the latter might just be too short to make a stable (non-clamped) complex. The experiment is consistent with a clamped complex but does not formally show it. Or perhaps instead of Benzonase treatment, treat the complex with a restriction enzyme after the fact and show it disassembles. This data can be added as a supplemental figure so the authors don't need to repeat the original and redraw the figure. If not performed, then the language should reflect that the experiment does not prove topological constraints.

3) More critically, reviewers agree that the single MAJOR issue is the ChIP-seq data in Figure 8A. We all expected an expanded profile around parS sites with ParB[E102A] and that is clearly not happening. Also, the authors highlight the upstream extended signal that is amplified in the mutant and largely absent in WT, but this could just be a confounding interaction with the par operon or expression from the parB gene itself. All three reviewers appreciate the authors trying to bring their in vitro findings back into the cell, but it did not seem to work out as described in the text. Of note, the Gruber lab currently has a similar story on BioRxiv and their ChIP-seq data for their CTP-trap mutant of B.sub ParB shows an expansion of signal around the parS sites. As noted in our independent reviews, the authors need to re-interpret their ChIP-seq data, provide a better and more detailed explanation of these findings, and possibly tamper their claims.

Finally, please see the "Recommendations to the Authors" section from all three reviewers for suggestions to improve clarity and presentation before resubmission.

*Reviewer #1 (Recommendations for the authors):*

Ln 302-304 and Figure 8

The description and interpretation of the [E102A] ChIP-seq data, as having a more "extended profile", is not an entirely accurate. Figure 8A clearly shows that spread is only in one direction (upstream), and there is a massive signal around the parAB operon (particularly over the ParB gene) with [E102A] that is largely absent from the WT ChIP seq. This strong [E102A] signal over the parAB operon dissipates in both directions, and the spread from this specific location is described by the authors as [E102A]'s "extended profile". This result is striking but not discussed by the authors. The authors need to expand their description and interpretation of the ChIP data as it relates to this region, and speculate as to what they think this upstream signal around the parAB operon may represent.

Ln 316-317 and Figure 9

Given the flexibility of the N-terminal domain, do the authors think it possible that this domain flexibility plays a role in ParB's ability to not only close the ring, but also associate with other ParB dimers and/or with ParA? Given that the ParA interaction interface is on this flexible domain, I think the data here interfaces quite nicely with the BioRxiv preprint from the Mizuuchi group (already cited in the paper) that was also recently reviewed by *eLife*: https://www.biorxiv.org/content/10.1101/2021.01.24.427996v1. In particular: They find ParB can bind to ParA either using the two protomers of a single dimer or two protomers from distinct dimers. The former occurs in the absence of ligands, the latter upon addition of either CTP or parS, thus presumably corresponding to the state of ParB found in the cells near a parS site. The discussion would benefit from the authors putting their findings in the context of other ParB interactions shown to be required for chromosome segregation – ParA interaction and ParB dimer oligomerization.

*Reviewer #2 (Recommendations for the authors):*

1. The crosslinking at L224 supports the idea that DNA has left the DNA binding domain after clamp closure, consistent with the structural clashes observed. Where is L224 in the DNA binding region? The cartoon in Figure 5 places it at the C-terminal side of the DBD but it would help if the position of L224 was indicated on one of the structures, eg Figure 1B, in addition to the cartoons. Why is there a helix 10 in the structure (Figure 1) but not in the secondary structure of Figure 2-supp2? By my estimation L224 is in helix 10 but this is confusing.

2. Figure 8: The ChIPseq profile of E102A is confusing and warrants more explanation. The authors state it is "notably more extended than the profile of" wild-type ParB. The shaded area to which they refer does not center on a parS site. What is the explanation for this pattern? That the profile is more extended also predicts that the other parS peaks would be more extended, which does not appear to be the case. Why? If the peaks were normalized to the same height, would E102A peaks look broader?

3. I like the alanine scanning approach to target the residues that interact with CTP, in part because it is a more comprehensive survey of the CTP binding site, which is poorly understood compared to sites that bind ATP or GTP. This allowed them to identify E102A as a mutation that created a CTP dependent clamp with diminished dissociation rate because it could not hydrolyze CTP. But the analysis also created two other classes of mutants (pg 8). Although the authors would likely argue that their analysis is beyond the scope of the current study, it would add important insight to the story if the authors discussed or proposed explanations for the behavior of each class in the Discussion. For example, why do class II, which bind but do not hydrolyse CTP, behave differently than E102A? Based on their location could they be permanently closed?

4. Figure 5: There should be a linearized plasmid control. The authors use the binding to circular DNA as evidence that ParB is topologically constrained, compared to a 22 bp linear DNA substrate. But the latter might just be too short to make a stable (non-clamped) complex. The experiment is consistent with a clamped complex but does not formally show it. Or perhaps instead of Benzonase treatment, treat the complex with a restriction enzyme after the fact and show it disassembles.

*Reviewer #3 (Recommendations for the authors):*

The data on the recycling mechanism upon CTP hydrolysis presented in Figure 8A are not sufficiently convincing to conclude that without CTP hydrolysis (variant E102A) the clamped-ParB diffuse longer away from parS than wt ParB. Especially, the most important signal, which strongly extend the ParB signature over DNA on the left side, seems independent of parS loading; indeed, the peak is away from the first and weak parS site. It rather corresponds to the location of the parB gene, which suggests that the signal detected here does not correspond to sliding but rather to ParB synthesis. Therefore, this is against the author's conclusion that preventing CTP hydrolysis extend ParB sliding. In addition, the signal over the parB gene is increased 4-5 fold compare to wt while the parS-specific signals are decreased about 3-fold. This may prevent a direct comparison of the "spreading" signature. Also, if ParB (E102A) is trapped longer on DNA, one would expect that the ParB signal should be higher in between parS sites that are close together such as in between parS3-4-5-6. The authors should comment on this lack of increase, and discuss these in regard of the sliding model relatively to other assembly model.

---

## [Author Response]

Essential revisions:All three reviewers agree that the work is exciting, high quality, will be of broad interest, and should be published in eLife. However, in the "Recommendations to the Authors" there are two control experiments that all three reviewers agree are simple to perform and required. Also, all three reviewers had concerns regarding the ChIP-seq data that need to be addressed prior to publication.1) From Reviewer 1: A control experiment where the authors pre-treat ParB with cross-linker BEFORE addition of DNA substrate to show that premature and irreversible closing of the ring prevents interactions with parS as well as spreading/sliding. The pre-crosslinking might alter protein conformation in other ways (besides closing a gate) that might destroy the DNA binding site (so there are caveats). Nevertheless if crosslinking did not prevent DNA binding, that would affect the overall story.

A premature closing of the ParB clamp, before being exposed to *parS* DNA, indeed prevents nucleation at *parS* as well as spreading/sliding. This observation was previously reported for *C. crescentus* ParB (Figure 6 of Jalal et al., *eLife* 2020). There, a prolonged (30-60min) pre-incubation of ParB (WT) with CTPɣS eliminated most of ParB binding to and sliding from *parS*. CTPɣS, even in the absence of *parS* DNA, gradually converted apo-ParB from an open to a closed protein clamp (Jalal et al., 2020; Soh et al., 2019). If not already bound on DNA, the now inaccessible DNA-binding domain of ParB cannot bind *parS* to nucleate and to subsequently slide to accumulate on DNA.

In the current manuscript, we have repeated the same experiment but with a crosslinkable ParB (Q35C), as requested by the reviewer, and observed the same behavior in the BLI assay (now reported in Figure 5—figure supplement 4B, 0 min vs. 30 min pre-incubation with CTPɣS).

We have also pre-crosslinked the closed-clamp form of ParB (Q35C) and subsequently used the crosslinked proteins in the same BLI assay (Figure 5—figure supplement 4B). To do so, purified ParB (Q35C) was incubated with CTPɣS for 30 min, then 1 mM BMOE was added to crosslink the N-terminal domains of ParB (Q35C) together. The crosslinking reaction was quenched with 1 mM DTT and ParB (Q35C) was purified away from excess BMOE and DTT using a Zeba buffer exchange column (See Materials and methods). Approx. 70% ParB (Q35C) was in a crosslinked form, consistent with a previous report (Jalal et al., 2020). Note that CTPɣS was required to convert apo-ParB (Q35C) to a closed clamp form before crosslinking. Without CTPɣS, most apo-ParB (E35C) is in an open clamp form (Jalal et al., 2020).

We have now added a sentence to the Results section to bring this important control to the attention of readers:

“This is also consistent with experiments that showed a premature and irreversible closing of ParB clamps, achieved either by an extended preincubation with CTPɣS (Jalal et al., 2020a and Figure 5—figure supplement 4B) or by pre-crosslinking a closed clamp form of ParB (Figure 5—figure supplement 4C), prevented nucleation at parS and DNA entrapment.”

2) From Reviewer 2: A linearized plasmid control. The authors use the binding to circular DNA as evidence that ParB is topologically constrained, compared to a 22 bp linear DNA substrate. But the latter might just be too short to make a stable (non-clamped) complex. The experiment is consistent with a clamped complex but does not formally show it. Or perhaps instead of Benzonase treatment, treat the complex with a restriction enzyme after the fact and show it disassembles. This data can be added as a supplemental figure so the authors don't need to repeat the original and redraw the figure. If not performed, then the language should reflect that the experiment does not prove topological constraints.

The reviewer raised important points and we have now performed additional controls with a linearized 3-kb *parS* DNA (reported in Figure 5—figure supplement 1).

We performed double BMOE crosslinking assays of dual-cysteine ParB variants + (circular/linearized) *parS*-containing plasmid DNA ± CTP. Purified ParB (Q35C I304C), ParB (L224C I304C), and ParB (Q35C L224C) were used for reactions in lanes 1-4, 5-8, and 9-12, respectively (Figure 5—figure supplement 1). Different DNA were employed in crosslinking reactions: a circular 3-kb *parS* plasmid (3 kb *parS* cir) or a 3-kb *parS* plasmid that had been linearized at an unique HindIII site by HindIII restriction enzyme (3 kb *parS* linear).

The high molecular weight (HMW) smear near the top of the polyacrylamide gel was observed only when ParB (Q35C I304C) or ParB (L224C I304C) was incubated with CTP and a circular *parS* plasmid (lanes 2 and 6, solid lines and asterisks, Figure 5—figure supplement 1), but not when a linearized *parS* plasmid was used (lanes 4 and 8, dashed lines and asterisks) or when CTP was omitted (lanes 1 and 5). ParB (Q35C L224C), as expected, did not produce a HMW smear even in the presence of CTP and a circular *parS* plasmid (see also Figure 5—figure supplement 2).

Overall, these results are consistent with a closed ParB clamp entrapping a topologically closed *parS* DNA, specifically within the DBD-CTD compartment. We have now revised the Results and Materials and Methods sections to describe these experiments.

3) More critically, reviewers agree that the single MAJOR issue is the ChIP-seq data in Figure 8A. We all expected an expanded profile around parS sites with ParB[E102A] and that is clearly not happening. Also, the authors highlight the upstream extended signal that is amplified in the mutant and largely absent in WT, but this could just be a confounding interaction with the par operon or expression from the parB gene itself. All three reviewers appreciate the authors trying to bring their in vitro findings back into the cell, but it did not seem to work out as described in the text. Of note, the Gruber lab currently has a similar story on BioRxiv and their ChIP-seq data for their CTP-trap mutant of B.sub ParB shows an expansion of signal around the parS sites. As noted in our independent reviews, the authors need to re-interpret their ChIP-seq data, provide a better and more detailed explanation of these findings, and possibly tamper their claims.

We agree entirely with all three reviewers and thank them for pointing this out. We have now revised the text and performed extra experiement to address the concern from the reviewers.

First, we have now described the ChIP-seq profile of ParB (E102A) in *C. crescentus* more accurately. Specifically, we removed this sentence “…the ChIP-seq profile of FLAG-ParB (E102A) … is notably more extended than the profile of FLAG-ParB (WT)…”. We highlighted the caveat associated with the instability of FLAG-ParB (E102A) protein in *C. crescentus* and cautioned against interpreting the roles of CTP hydrolysis in native *C. crescentus* host, both in the Results and the Discussion sections. We wrote:

“By contrast, the ChIP-seq profile of FLAG-ParB (E102A) is significantly reduced in height but has an extra peak over the parB coding sequence (Figure 8A, asterisk). […] Again, due to the caveat of a lower ParB (E102A) protein level in *C. crescentus* (Figure 8—figure supplement 1A), we could not reliably link the in vitro properties of ParB (E102A) to its behaviors in the native host.”

Second, to overcome the caveat of protein instability, we instead investigated the spreading of ParB (WT) vs. ParB (E102A) from *parS* by analysing the *C. crescentus* ParB/*parS* system in *E. coli*. *E. coli* does not possess a ParA/ParB homolog nor a *parS*-like sequence, thus it serves as a suitable heterologous host. *C. crescentus parS* sites 3 and 4 were engineered onto the *E. coli* chromosome at the *ygcE* locus (Figure 8B). CFP-tagged ParB (WT/E102A) was expressed from a leaky lactose promoter (P*_lac_*, no IPTG was added) on a medium-copy-number plasmid. CFP-ParB (WT/E102A) was produced at the same level, as judged by an immunoblot (Figure 8—figure supplement 1B).

We observed by ChIP-seq that CFP-ParB (WT) in an *E. coli* host spreads asymmetrically ~5 kb around *parS* sites. By contrast, the shape of the ParB (E102A) distribution was clearly different from that of ParB (WT); the profile was further expanded to both neighboring sides of *parS* (covering in total ~26 kb) at the expense of the enrichment at *parS* itself (Figure 8B). The more excessive spreading of ParB (E102A) might suggest that this variant, in the absence of CTP hydrolysis, persisted and perhaps slid further away from the loading site *parS* in *E. coli*. The reduced enrichment of ParB (E102A) at *parS* itself (Figure 8B) might be due to reduced cytoplasmic ParB (E102A) available to re-nucleate at *parS* and/or due to stably entrapped ParB (E102A) sterically hindering further nucleation events. We further noted that the ChIP-seq profile of CFP-ParB (E102A) in *E. coli* is highly asymmetrical, with more enrichment in the +2905-2911kb region than the +2885-2899kb region (shaded areas, Figure 8B). The asymmetrical spreading is possibly due to an impediment in one direction by roadblocks such as RNA polymerases or DNA-bound proteins, which have been shown previously to be able to interfere with ParB spreading (Balaguer et al., 2021; Breier and Grossman, 2007; Jalal et al., 2020; Murray et al., 2006; Rodionov et al., 1999; Soh et al., 2019). We also quantified the fluorescence intensity of CFP-ParB (WT/E102A) foci, and found a higher CFP signal for CFP-ParB (E102A) foci than that of CFP-ParB (WT) (Figure 8—figure supplement 3). This is also consistent with a scenario where CFP-ParB (E102A) occupies a more extended genomic area in our *E. coli* host. Altogether, at least in the heterologous *E. coli* host, the “clamp-locked” phenotype of ParB (E102A) implies a possible role of CTP hydrolysis and/or the release of hydrolytic products in re-opening wild-type ParB clamp to release DNA and to recycle ParB.

Lastly, we also noted that two preprints from the Gruber and the Thanbichler labs also reported the same phenomenon that the enrichment/height at *parS* itself is reduced but the enriched area around *parS* is expanded in the “clamp-locked” mutants in *B. subtilis* and *M. xanthus* (Antar et al., 2021; Osorio-Valeriano et al., 2021).

Finally, please see the "Recommendations to the Authors" section from all three reviewers for suggestions to improve clarity and presentation before resubmission.

Detailed responses to the specific points that reviewers have raised are given below:

Reviewer #1 (Recommendations for the authors):Ln 302-304 and Figure 8The description and interpretation of the [E102A] ChIP-seq data, as having a more "extended profile", is not an entirely accurate. Figure 8A clearly shows that spread is only in one direction (upstream), and there is a massive signal around the parAB operon (particularly over the ParB gene) with [E102A] that is largely absent from the WT ChIP seq. This strong [E102A] signal over the parAB operon dissipates in both directions, and the spread from this specific location is described by the authors as [E102A]'s "extended profile". This result is striking but not discussed by the authors. The authors need to expand their description and interpretation of the ChIP data as it relates to this region, and speculate as to what they think this upstream signal around the parAB operon may represent.

Please see my response to point 3 in the Essential Revisions for Authors.

Ln 316-317 and Figure 9Given the flexibility of the N-terminal domain, do the authors think it possible that this domain flexibility plays a role in ParB's ability to not only close the ring, but also associate with other ParB dimers and/or with ParA? Given that the ParA interaction interface is on this flexible domain, I think the data here interfaces quite nicely with the BioRxiv preprint from the Mizuuchi group (already cited in the paper) that was also recently reviewed by eLife: https://www.biorxiv.org/content/10.1101/2021.01.24.427996v1. In particular: They find ParB can bind to ParA either using the two protomers of a single dimer or two protomers from distinct dimers. The former occurs in the absence of ligands, the latter upon addition of either CTP or parS, thus presumably corresponding to the state of ParB found in the cells near a parS site. The discussion would benefit from the authors putting their findings in the context of other ParB interactions shown to be required for chromosome segregation – ParA interaction and ParB dimer oligomerization.

These are excellent points and we have expanded the Discussion as followed:

“Recently, an F-plasmid ParB was shown to form biomolecular condensates in vivo that might bridge distal ParB_F_ dimers together (Guilhas et al., 2020; Walter et al., 2020). […] Future work will provide important insights to better understand the mechanism of ParA-directed DNA segregation.”

We refrain from over-speculating here because we do not know enough and have not performed any experiments with phase separation or ParA in vivo. We, however, highlighted a few questions which we think are very important and hopefully they will motivate further investigation from multiple labs in our field.

Reviewer #2 (Recommendations for the authors):1. The crosslinking at L224 supports the idea that DNA has left the DNA binding domain after clamp closure, consistent with the structural clashes observed. Where is L224 in the DNA binding region? The cartoon in Figure 5 places it at the C-terminal side of the DBD but it would help if the position of L224 was indicated on one of the structures, eg Figure 1B, in addition to the cartoons. Why is there a helix 10 in the structure (Figure 1) but not in the secondary structure of Figure 2-supp2? By my estimation L224 is in helix 10 but this is confusing.

Residue L224 locates in the loop that connects helix 9 and helix 10 of the DNA-binding domain (DBD). So the reviewer is correct that L224 positions towards the far C-terminal side of the DBD. We have now indicated the positions of L224 in Figure 1B.

The reviewer is correct that helix 10 can be seen in the ParB∆CTD-*parS* structure (Figure 1B) but not in the CTPɣS-bound structure. This is because of the poor electron density for helix 10 in the CTPɣS-bound structure, which prevented us from modeling in this last helix of the DBD. We have now mentioned this in the lenged of Figure 2 (that describes the CTPɣS-bound structure) to make this clear to readers.

2. Figure 8: The ChIPseq profile of E102A is confusing and warrants more explanation. The authors state it is "notably more extended than the profile of" wild-type ParB. The shaded area to which they refer does not center on a parS site. What is the explanation for this pattern? That the profile is more extended also predicts that the other parS peaks would be more extended, which does not appear to be the case. Why? If the peaks were normalized to the same height, would E102A peaks look broader?

Please see my response to point 3 in the Essential Revisions for Authors. Also, if the peaks were normalized to the same height, E102A peaks (in the ChIP-seq profiles for *C. crescentus*) are not broader either.

3. I like the alanine scanning approach to target the residues that interact with CTP, in part because it is a more comprehensive survey of the CTP binding site, which is poorly understood compared to sites that bind ATP or GTP. This allowed them to identify E102A as a mutation that created a CTP dependent clamp with diminished dissociation rate because it could not hydrolyze CTP. But the analysis also created two other classes of mutants (pg 8). Although the authors would likely argue that their analysis is beyond the scope of the current study, it would add important insight to the story if the authors discussed or proposed explanations for the behavior of each class in the Discussion. For example, why do class II, which bind but do not hydrolyse CTP, behave differently than E102A? Based on their location could they be permanently closed?

We noted that Class II mutants ParB (Q58A) and ParB (E135A) already had an elevated crosslinking efficiency even in the absence of CTP (Figure 6C). This premature clamp closing might indeed result in a less than wild-type level of DNA entrapment (Figure 6D). The reason the clamp closes prematurely for Class II mutants is not yet clear, and cannot easily be rationalized based on our current crystal structures, especially for the Q85A substitution. We wish to investiagate this class of mutants more closely in the near future. We hope the reviewer will understand.

4. Figure 5: There should be a linearized plasmid control. The authors use the binding to circular DNA as evidence that ParB is topologically constrained, compared to a 22 bp linear DNA substrate. But the latter might just be too short to make a stable (non-clamped) complex. The experiment is consistent with a clamped complex but does not formally show it. Or perhaps instead of Benzonase treatment, treat the complex with a restriction enzyme after the fact and show it disassembles.

Please see my response to point 2 in the Essential Revisions for Authors.

References:

Antar H, Soh Y-M, Zamuer S, Bock FP, Anchimiuk A, Rios PDL, Gruber S. 2021. Relief of ParB autoinhibition by parS DNA catalysis and ParB recycling by CTP hydrolysis promote bacterial centromere assembly. bioRxiv 2021.05.05.442573. doi:10.1101/2021.05.05.442573

Balaguer F de A, Aicart-Ramos C, Fisher GL, de Bragança S, Martin-Cuevas EM, Pastrana CL, Dillingham MS, Moreno-Herrero F. 2021. CTP promotes efficient ParB-dependent DNA condensation by facilitating one-dimensional diffusion from parS. *eLife* 10:e67554. doi:10.7554/*eLife*.67554

Breier AM, Grossman AD. 2007. Whole-genome analysis of the chromosome partitioning and sporulation protein Spo0J (ParB) reveals spreading and origin-distal sites on the *Bacillus subtilis* chromosome. Molecular Microbiology 64:703–718. doi:10.1111/j.1365-2958.2007.05690.x

Guilhas B, Walter J-C, Rech J, David G, Walliser NO, Palmeri J, Mathieu-Demaziere C, Parmeggiani A, Bouet J-Y, Le Gall A, Nollmann M. 2020. ATP-Driven Separation of Liquid Phase Condensates in Bacteria. Mol Cell 79:293-303.e4. doi:10.1016/j.molcel.2020.06.034

Jalal AS, Tran NT, Le TB. 2020. ParB spreading on DNA requires cytidine triphosphate in vitro. *eLife* 9:e53515. doi:10.7554/*eLife*.53515

Murray H, Ferreira H, Errington J. 2006. The bacterial chromosome segregation protein Spo0J spreads along DNA from parS nucleation sites. Molecular Microbiology 61:1352–1361. doi:10.1111/j.1365-2958.2006.05316.x

Osorio-Valeriano M, Altegoer F, Das CK, Steinchen W, Panis G, Connolley L, Giacomelli G, Feddersen H, Corrales-Guerrero L, Giammarinaro P, Hanßmann J, Bramkamp M, Viollier PH, Murray S, Schäfer LV, Bange G, Thanbichler M. 2021. The CTPase activity of ParB acts as a timing mechanism to control the dynamics and function of prokaryotic DNA partition complexes. bioRxiv 2021.05.05.442810. doi:10.1101/2021.05.05.442810

Rodionov O, Lobocka M, Yarmolinsky M. 1999. Silencing of genes flanking the P1 plasmid centromere. Science 283:546–549. doi:10.1126/science.283.5401.546

Sanchez A, Cattoni DI, Walter J-C, Rech J, Parmeggiani A, Nollmann M, Bouet J-Y. 2015. Stochastic Self-Assembly of ParB Proteins Builds the Bacterial DNA Segregation Apparatus. Cell Syst 1:163–173. doi:10.1016/j.cels.2015.07.013

Soh Y-M, Davidson IF, Zamuner S, Basquin J, Bock FP, Taschner M, Veening J-W, De Los Rios P, Peters J-M, Gruber S. 2019. Self-organization of parS centromeres by the ParB CTP hydrolase. Science 366:1129–1133. doi:10.1126/science.aay3965

Taylor JA, Seol Y, Neuman KC, Mizuuchi K. 2021. CTP and parS control ParB partition complex dynamics and ParA-ATPase activation for ParABS-mediated DNA partitioning. bioRxiv 2021.01.24.427996. doi:10.1101/2021.01.24.427996

Walter J-C, Rech J, Walliser N-O, Dorignac J, Geniet F, Palmeri J, Parmeggiani A, Bouet J-Y. 2020. Physical Modeling of a Sliding Clamp Mechanism for the Spreading of ParB at Short Genomic Distance from Bacterial Centromere Sites. iScience 23:101861. doi:10.1016/j.isci.2020.101861